# THE BENEFIT OF BEING BAYESIAN IN ONLINE CONFORMAL PREDICTION

## ABSTRACT

Based on the framework of Conformal Prediction (CP), we study the online construction of valid confidence sets given a black-box machine learning model. By converting the target confidence levels into quantile levels, the problem can be reduced to predicting the quantiles (in hindsight) of a sequentially revealed data sequence. Two very different approaches have been studied previously:

- *Direct approach:* Assuming the data sequence is iid or exchangeable, one could maintain the empirical distribution of the observed data as an algorithmic belief, and directly predict its quantiles.
- *Indirect approach:* As statistical assumptions often do not hold in practice, a recent trend is to consider the adversarial setting and apply first-order online optimization to moving quantile losses (Gibbs & Candès, 2021). It requires knowing the target quantile level beforehand, and suffers from a monotonicity issue on the obtained confidence sets, due to the associated loss linearization.

This paper presents a novel Bayesian CP framework that combines their strengths. Without any statistical assumption, it is able to both

- answer multiple arbitrary confidence level queries online, with provably low regret; and
- overcome the monotonicity issue suffered by first-order optimization baselines, due to being "data-centric" rather than "iterate-centric".

In addition, it can adapt to an iid environment with the correct coverage probability guarantee.

From a technical perspective, our key idea is to regularize the algorithmic belief of the above direct approach by a Bayesian prior, which "robustifies" it by simulating a non-linearized *Follow the Regularized Leader* (FTRL) algorithm on the output. For statisticians, this can be regarded as an online adversarial view of Bayesian inference. Importantly, the proposed belief update backbone is shared by prediction heads targeting different confidence levels, bringing practical benefits analogous to the recently proposed concept of *U-calibration* (Kleinberg et al., 2023).

## 1 INTRODUCTION

Modern machine learning (ML) models are better at point prediction compared to probabilistic prediction. For example, when given an image classification task, they are better at responding "*this image is most likely a white cat*", rather than "*I'm 90% sure this image is an animal, 60% sure it's a cat, and 30% sure it's a white cat*". For downstream users, the more nuanced probabilistic predictions are often important for risk assessment. The challenge, however, lies in aligning the model's own uncertainty evaluation with its actual performance in the real world.

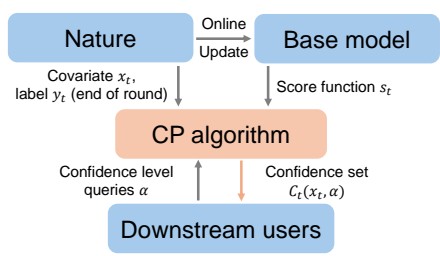

Figure 1: The CP interaction protocol.

*Conformal prediction* (CP) (Vovk et al., 2005) has recently emerged as a premier framework to address this challenge, as it blends the empirical strength of modern ML with the theoretical soundness of traditional statistical methods. As illustrated in Figure 1, CP algorithms make *confidence set*

*predictions* on the label space, by sequentially interacting with three other parties: the *nature* (i.e., the data stream), a *black-box ML model*, and *downstream users*. In each (the $t$-th) round,

1. We, as the CP algorithm, observe a *target covariate* $x_t \in \mathcal{X}$ from the nature, and a *score function* $s_t : \mathcal{X} \times \mathcal{Y} \to [0, R]$ generated by a black-box ML model BASE.

2. The downstream users select a finite set of *confidence level queries*, $A_t \subset [0, 1]$.

3. Given each $\alpha \in A_t$, we predict a *score threshold* $r_t(x_t, \alpha)$ based on existing observations, which leads to a *confidence set*[1]

$$\mathcal{C}_t(x_t, \alpha) = \{y \in \mathcal{Y} : s_t(x_t, y) \leq r_t(x_t, \alpha)\}. \tag{1}$$

4. Nature reveals the *ground truth label* $y_t \in \mathcal{Y}$ and the *true score* $r_t^* := s_t(x_t, y_t)$ to us.

5. The $(x_t, y_t)$ pair is passed to BASE, which it optionally uses to generate the score function $s_{t+1}$.

By sequentially evaluating BASE on the target data, we generate better score thresholds that "correct" the uncertainty evaluation from BASE itself. As a concrete example, one could imagine BASE being a trained image classifier, and the user being a wildlife conservation organization that uses BASE to monitor endangered species. Generating a plethora of informative confidence sets would enable the user to have a more accurate understanding of the species at risk.

Our goal is thus clear in a very broad sense – predicting confidence sets with *guaranteed validity*. Say if a user queries the confidence level $\alpha = 90\%$, then our CP algorithm needs to provide certain quantitative evidence that incentivizes the user to treat $\mathcal{C}_t(x_t, \alpha)$ as the 90% confidence set about the true label $y_t$. While solutions are well-known in various statistical settings, the present work is about designing better CP algorithms without any statistical assumption at all.

## 1.1 BACKGROUND

To introduce the necessary background, we start from the simplest case: within a time horizon $T$, the true scores $r_{1:T}^*$ are iid samples of a random variable $X$ with strictly positive density. This can happen if the target data $(x_t, y_t)$ for different $t$ are iid, and BASE is fixed (i.e., Step 5 is skipped). Further suppose the $\alpha$-*quantile* of $X$,[2] denoted by $q_\alpha(X)$, is known, then a natural strategy is to predict $r_t(x_t, \alpha) = q_\alpha(X)$. This ensures that the *coverage condition* $y_t \in \mathcal{C}_t(x_t, \alpha)$ holds with probability exactly $\alpha$. That is, the confidence set prediction is valid in a strong probabilistic sense.

Although the assumptions are clearly unrealistic, this example illustrates a central principle of CP: the predicted score threshold $r_t(x_t, \alpha)$ should ideally be the $\alpha$-quantile of *some* distribution of $r_{1:T}^*$. A key challenge of CP is thus generalizing this principle to more realistic settings, as described below.

- *Direct approach:* Still assuming the sequence $r_{1:T}^*$ is iid but the population quantile $q_\alpha(X)$ is unknown, we could instead estimate $q_\alpha(X)$ on the fly. Specifically, our algorithm maintains the empirical distribution of $r_{1:t-1}^*$, denoted by $P_t = \bar{P}(r_{1:t-1}^*)$, as an *algorithmic belief* about the unknown distribution of $X$. Then, when queried with any confidence level $\alpha$, it "post-processes" the belief by setting $r_t(x_t, \alpha) = q_\alpha(P_t)$. This is equivalent to *Empirical Risk Minimization* (ERM) with the *quantile loss* $l_\alpha(r, r^*) := (\mathbf{1}[r \geq r^*] - \alpha)(r - r^*)$, i.e.,

$$r_t(x_t, \alpha) = q_\alpha(P_t) \in \underset{r \in [0, R]}{\arg\min} \sum_{i=1}^{t-1} l_\alpha(r, r_i^*). \tag{2}$$

A standard improvement called *Split Conformal* (Papadopoulos et al., 2002) sets $r_t(x_t, \alpha) = q_{\alpha+o(1)}(P_t)$, where the $o(1)$ offset (wrt $t \to \infty$) ensures that even under a relaxation of iid called *exchangeability*, a suitable notion of coverage probability is lower bounded by $\alpha$ (Roth, 2022).

- *Indirect approach:* Since statistical assumptions often do not hold in practice, a recent trend (Gibbs & Candès, 2021) is to remove all statistical assumptions, and instead estimate the empirical quantile of $r_{1:T}^*$ using first-order optimization algorithms from *adversarial online learning* (Hazan, 2023;

---

[1]Without loss of generality, we assume the score function $s_t(x, y)$ is *negatively oriented*: smaller means the ML model is more certain that the candidate label $y$ is the true one. See Appendix A for an example.

[2]For the readers' reference, the $\alpha$-quantile of a real random variable $X$ is defined as $q_\alpha(X) := \min\{x : \mathbb{P}(X \leq x) \geq \alpha\}$.

Orabona, 2023). Taking gradient descent for example, such an approach amounts to picking an initialization $r_1(x_1, \alpha) \in [0, R]$ and following with the incremental update

$$r_{t+1}(x_{t+1}, \alpha) = r_t(x_t, \alpha) - \eta_t \partial l_\alpha(r_t(x_t, \alpha), r_t^*), \tag{3}$$

where $\eta_t > 0$ is the *learning rate*, and $\partial l_\alpha(r, r^*)$ can be any subgradient of the quantile loss $l_\alpha$ with respect to the first argument. Due to the absence of probability, alternative performance metrics have to be considered, such as the *post-hoc coverage frequency* and the *regret*.

How do these two approaches compare? Although first-order optimization does not need statistical assumptions, it requires being "iterate-centric" rather than "data-centric": one needs to fix a single confidence level $\alpha$ beforehand, and the predicted threshold $r_t(x_t, \alpha)$ depends on how previous predictions compare to the true scores $r_{1:t-1}^*$, rather than just $r_{1:t-1}^*$ itself. This leads to a critical monotonicity issue regarding the obtained confidence sets:

- As demonstrated in Section 2, two copies of an algorithm with $\alpha_1 < \alpha_2$ can output $\mathcal{C}_t(x_t, \alpha_2) \subsetneq \mathcal{C}_t(x_t, \alpha_1)$, even if the initializations are the same. That is, the higher-confidence set is strictly smaller, violating the monotonicity of probability measures.

In contrast, the direct ERM approach does not suffer from this issue. The problem is that being equivalent to *Follow the Leader* (FTL) in online learning, it is well-known that ERM can suffer the vacuous $\Omega(T)$ regret on adversarial quantile losses. This motivates an important question:

Can we design an adaptive CP algorithm that enjoys the best of both worlds?

## 1.2 OUR RESULT

This paper presents a novel *Bayesian* approach that combines several strengths of previous attempts.

- Just like the ERM approach, it can answer multiple arbitrary confidence level queries online.
- Without any statistical assumption, and with just the uniform prior, it guarantees the optimal "frequentist" regret bound

$$\text{Regret}_T(\alpha) := \sum_{t=1}^{T} l_\alpha(r_t(x_t, \alpha), r_t^*) - \sum_{t=1}^{T} l_\alpha(q_\alpha(r_{1:T}^*), r_t^*) = O(R\sqrt{T}),$$

simultaneously for all time horizon $T$, true score sequence $r_{1:T}^*$, and confidence level $\alpha \in [0, 1]$. Notice that the comparator $q_\alpha(r_{1:T}^*)$ would be a natural fixed prediction had one known the empirical distribution of the true score sequence $r_{1:T}^*$ beforehand.
- Unlike first-order optimization baselines, it does not suffer from the aforementioned monotonicity issue, due to being "data-centric" rather than "iterate-centric".
- Under the iid assumption, it achieves almost the same guarantees, including the *dataset-conditional coverage probability* and the *excess quantile risk*, as the ERM baseline.

A particular benefit of these strengths can be viewed from the perspective of adaptivity, as it is important to have an algorithm with performance guarantees in both iid and adversarial environments. Think about this: in many practical applications of CP one has to apply the algorithm without knowing the nature of environment in advance. In such cases it is impossible to say something like "we'll apply Split CP if the data sequence is exchangeable, and ACI otherwise"; instead, an ideal algorithm needs to work well under all data-generation mechanisms. Our algorithm is equipped with adaptivity of this type, alongside its ability to support multiple confidence level queries "coreherently".

From a technical perspective, our algorithm is a simple Bayesian modification of the ERM approach: instead of setting the algorithmic belief as the empirical distribution of the past, $P_t = \bar{P}(r_{1:t-1}^*)$, we set it as the convex combination

$$P_t = \lambda_t P_0 + (1 - \lambda_t)\bar{P}(r_{1:t-1}^*),$$

where $P_0$ is a prior, and $\lambda_t \in [0, 1]$ is a hyperparameter. The key observation is that this *Bayesian distribution estimator* leads to *downstream regularization*: the associated score threshold prediction $r_t(x_t, \alpha) = q_\alpha(P_t)$ is equivalent to the output of a non-linearized *Follow the Regularized Leader* (FTRL) algorithm, from which the regret bound naturally follows.

### 1.3 RELATED WORK

**Online CP** Adversarial online CP was first studied by Gibbs & Candès (2021). It was shown that gradient descent with constant learning rate can guarantee low *coverage frequency error*, i.e.,

$$\left| \alpha - T^{-1} \sum_{t=1}^{T} \mathbf{1}[r_t^* \le r_t(x_t, \alpha)] \right| = o(1), \tag{4}$$

as well as its sliding-window analogues. Later, Bastani et al. (2022) demonstrated a weakness of this performance metric: one could trivially satisfy this coverage frequency bound by predicting a data-independent alternation between the empty set and the entire label space. To rule out such cases, the typical solution is to consider an additional performance metric, such as the regret (Bhatnagar et al., 2023; Gibbs & Candès, 2024; Zhang et al., 2024) and the *multi-calibrated* coverage frequency (Bastani et al., 2022). Under the additional iid assumption, Angelopoulos et al. (2024) studied the asymptotic coverage probability achieved by gradient descent.

The present work focuses on regret minimization, as we believe such a perspective offers advantages *even over simultaneously bounding the regret and the coverage frequency error* (since loss linearization is not necessary anymore). See Section 3 for a thorough discussion.

**Adversarial Bayes** Making sequential Bayesian methods "adversarially robust" is closely related to the classical *Follow the Perturbed Leader* (FTPL) algorithm in online learning (Kalai & Vempala, 2005). Notable examples of FTPL include *Thompson sampling*, a prevalent Bayesian approach for bandits and reinforcement learning (Thompson, 1933; Lattimore & Szepesvári, 2020; Xu & Zeevi, 2023), and *U-calibration* (Kleinberg et al., 2023; Luo et al., 2024), a recently proposed framework for loss-agnostic decision making. Despite being deterministic, our approach resembles the high level idea of U-calibration and a related concept called *omniprediction* (Gopalan et al., 2022; Garg et al., 2024). The connections and differences are discussed in Section 5.

Additional discussion of related works is deferred to Appendix A, including an independent topic called *Bayesian uncertainty quantification* which motivated CP in the first place.

### 1.4 NOTATION

This paper studies the *marginal* setting of CP, which means the threshold prediction $r_t(x_t, \alpha)$ will be independent of $x_t$; therefore we write it as $r_t(\alpha)$ for conciseness. For any symbol $x$, $x_{1:t}$ (e.g., $r_{1:t}^*$) represents the tuple $[x_1, \ldots, x_t]$. $\bar{P}(\cdot)$ denotes the empirical distribution of its input, and $q_\alpha(\cdot)$ denotes the $\alpha$-quantile. Our regret bound concerns the quantile (or pinball) loss defined as $l_\alpha(r, r^*) := (\mathbf{1}[r \ge r^*] - \alpha)(r - r^*)$. log denotes the natural logarithm.

## 2 THE NEED FOR MONOTONICITY

To begin with, we use a numerical experiment to elaborate a validity issue suffered by existing adversarial online CP algorithms: the predicted confidence sets can violate the monotonicity of probability measures. This has been overlooked in the literature, as all the existing approaches we are aware of require fixing a single target confidence level $\alpha$ at the beginning of the CP game.

Specifically, we consider two baselines, *Online Gradient Descent* (OGD) from (Gibbs & Candès, 2021), and *MultiValid Prediction* (MVP) from (Bastani et al., 2022). To enable multiple confidence level queries, we adopt the following nearest-neighbor routing on top of their independent copies.

1. Evenly discretize the $[0, 1]$ interval of possible confidence levels using a grid $\tilde{A}$.

2. For each $\tilde{\alpha} \in \tilde{A}$, maintain a "base" online CP algorithm targeting $\tilde{\alpha}$.

3. Given any queried $\alpha$, follow the base algorithm corresponding to its nearest neighbor in $\tilde{A}$.

The resulting algorithms are named as MultiOGD and MultiMVP respectively.

In the experiment, we fix $R = 1$. The true score sequence $r_{1:T}^*$ is sampled iid from the uniform distribution on $[0, 1]$, and we evaluate the thresholds $r_{1:T}(\alpha)$ predicted by different CP algorithms, under different $\alpha$ values. For each base OGD targeting $\tilde{\alpha}$, we use the standard learning rate $\eta_t = t^{-1/2}$, and initialize it with $r_1(\tilde{\alpha}) = \tilde{\alpha}$. The base MVP algorithms are all initialized at 0 following (Bastani

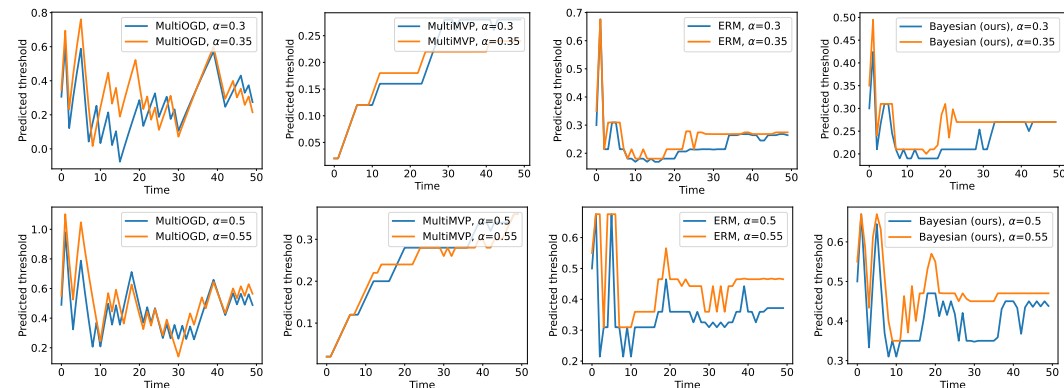

Figure 2: Evaluating the monotonicity of threshold predictions. Ideally the orange line should be always above the blue line, since the associated target confidence level is higher. Columns correspond to different algorithms; rows correspond to different confidence level pairs.

et al., 2022). The point is that the initialization of MultiOGD and MultiMVP cannot be the reason of any monotonicity violation. For comparison, we also test ERM as well as our Bayesian algorithm to be introduced in Section 3.

Our results are visualized in Figure 2. Ideally, in all the figures the orange line should be always above the blue line (i.e., the predicted confidence set due to Eq.(1) is larger), since the associated confidence level $\alpha$ is higher. Unlike ERM and our Bayesian algorithm, both MultiOGD and MultiMVP violate this property, which somewhat harms their trustworthiness to downstream users. We remark that although the data generation mechanism and the MultiMVP baseline are both randomized, a single random seed is used in this experiment to demonstrate the existence of the problem.

## 3 BAYESIAN ONLINE CONFORMAL PREDICTION

Given a sneak peek into our algorithm, now let us take a deeper dive. Our core algorithm (Algorithm 1) is perhaps the simplest one could think of. Setting the *Bayesian prior* as an arbitrary distribution $P_0$ on $[0, R]$ with strictly positive density function $p_0$, we update the algorithmic belief $P_t$ by mixing $P_0$ with the empirical distribution of the previous true scores, $\bar{P}(r^*_{1:t-1})$. This can be seen as regularizing the frequentist belief update $P_t = \bar{P}(r^*_{1:t-1})$, and the readers are referred to Section 5 for an interpretation of this procedure as *Bayesian distribution estimation*. Then, given each queried confidence level $\alpha$, the algorithm picks $r_t(\alpha) = q_\alpha(P_t)$ just like the ERM approach.

Note that the algorithmic belief $P_t$ does not depend on any specific $\alpha$, and different downstream users can select different $\alpha$ values online. By construction, it is clear that for any $\alpha_1 < \alpha_2$ we always have $r_t(\alpha_1) \leq r_t(\alpha_2)$.

---

**Algorithm 1** Online conformal prediction with regularized belief.

**Require:** Step sizes $\{\lambda_t\}_{t \in \mathbb{N}_+}$, where $\lambda_1 = 1$, and $0 < \lambda_t < 1$ for all $t \geq 2$. Bayesian prior $P_0$ with strictly positive density function $p_0$.

1: **for** $t = 1, 2, \ldots$ **do**
2:    Compute the empirical distribution $\bar{P}(r^*_{1:t-1})$, and set the algorithmic belief $P_t$ to

$$P_t = \lambda_t P_0 + (1 - \lambda_t)\bar{P}(r^*_{1:t-1}). \tag{5}$$

3:    **for** $\alpha \in A_t$ **do**
4:        Output the score threshold $r_t(\alpha) = q_\alpha(P_t)$.
5:    **end for**
6:    Observe the true score $r^*_t$.
7: **end for**

---

The most important idea of this paper is the following observation.

The Bayesian regularization on the algorithmic belief $P_t$ induces *downstream regularizations* on the predicted threshold $r_t(\alpha)$, which best-responds to $P_t$.

Concretely, with a base regularizer defined as $\psi(r) := \mathbb{E}_{r^* \sim P_0}[l_\alpha(r, r^*)]$, we characterize this observation by the following equivalence theorem.

**Theorem 1.** *For all $\alpha \in [0, 1]$, the output $r_t(\alpha)$ of Algorithm 1 satisfies $r_1(\alpha) = \arg\min_{r \in \mathbb{R}} \psi(r)$,*

$$r_t(\alpha) = \arg\min_{r \in \mathbb{R}} \left[ \frac{\lambda_t(t-1)}{1 - \lambda_t} \psi(r) + \sum_{i=1}^{t-1} l_\alpha(r, r_i^*) \right], \quad \forall t \geq 2. \tag{6}$$

*Specifically,*

* $\psi$ *is strongly convex with coefficient* $\inf_{r \in [0,R]} p_0(r)$*, if the latter is positive.*

* *If $P_0$ is the uniform distribution on $[0, R]$, then $\psi$ is a quadratic function centered at $\alpha R$,*

$$\psi(r) = \frac{1}{2R} r^2 - \alpha r + \frac{1}{2} \alpha R.$$

Theorem 1 shows that despite not knowing $\alpha$ at the beginning of the CP game, Algorithm 1 generates the same output $r_t(\alpha)$ as a non-linearized *Follow the Regularized Leader* (FTRL) algorithm on the quantile loss $l_\alpha$. Specifically, Eq.(6) can be compared to the FTL-equivalence of the iid-based approach, Eq.(2). The important difference is the additional regularizer $\psi(r)$.

To provide more context here: FTRL is a standard improvement of ERM / FTL in adversarial online learning, with better stability and worst-case performance on "difficult loss functions". Our analysis involves the non-linearized version of FTRL, which has previously received less attention than its linearized counterpart. This is largely due to computational reasons, since non-linearized FTRL has to solve a convex optimization subroutine in each round, whereas linearized FTRL admits closed-form solutions (Orabona, 2023, Chapter 7.3). From this perspective, a novelty of our result is showing that for a class of benign regularizers, non-linearized FTRL on quantile losses can be simulated by a simple and efficient Bayesian procedure.

From Theorem 1, we can then obtain the regret bound of Algorithm 1 using the standard FTRL analysis. In order to demonstrate the role of good priors, the strong convexity of the regularizer $\psi$ will be measured locally.

**Theorem 2.** *Let $\mu_{t,\alpha} := \inf\{p_0(r) : r_t(\alpha) \wedge r_t^* \leq r \leq r_t(\alpha) \vee r_t^*\}$. With the step size $\lambda_t = 1/\sqrt{t}$, Algorithm 1 guarantees*

$$\mathrm{Regret}_T(\alpha) := \sum_{t=1}^T l_\alpha(r_t(\alpha), r_t^*) - \sum_{t=1}^T l_\alpha(q_\alpha(r_{1:T}^*), r_t^*) = O\left( \psi(q_\alpha(r_{1:T}^*))\sqrt{T} + \sum_{t=1}^T \frac{1}{\mu_{t,\alpha}\sqrt{t}} \right), \tag{7}$$

*for all time horizon $T$, true score sequence $r_{1:T}^*$, and confidence level $\alpha \in [0, 1]$. Here, $O(\cdot)$ subsumes an absolute constant. Furthermore, if $P_0$ is the uniform distribution on $[0, R]$, then*

$$\mathrm{Regret}_T(\alpha) = O(R\sqrt{T}).$$

Let us interpret this regret bound. Suppose the time horizon $T$ and the empirical true score distribution $\bar{P}(r_{1:T}^*)$ are known beforehand (but the exact $r_{1:T}^*$ sequence is unknown), then for all $\alpha$, a very natural strategy is to predict $r_t(\alpha) = q_\alpha(r_{1:T}^*)$. Theorem 2 shows that without any statistical assumption, Algorithm 1 with the uniform prior asymptotically performs as well as this oracle in terms of the total quantile loss, and importantly, the $O(R\sqrt{T})$ regret bound is known to be tight (Hazan, 2023; Orabona, 2023). Existing first-order optimization baselines are equipped with regret bounds of a similar type (Bhatnagar et al., 2023; Gibbs & Candès, 2024; Zhang et al., 2024), but the difference is that they require knowing the confidence level $\alpha$ beforehand, whereas Algorithm 1 achieves low regret simultaneously for all $\alpha \in [0, 1]$.

**The role of good prior** A particular strength of Theorem 2 is that the $O(R\sqrt{T})$ regret bound only requires the simplest uniform prior. Nonetheless, if one has extra prior knowledge on the environment, picking a more sophisticated prior can indeed bring advantages. To see this, notice that the function

$\psi$ in Eq.(7) is minimized at $q_\alpha(P_0)$, therefore ideally we would aim for $P_0 \approx \bar{P}(r_{1:T}^*)$, which means $q_\alpha(P_0) \approx q_\alpha(r_{1:T}^*)$ for all $\alpha$. But unlike the discrete distribution $\bar{P}(r_{1:T}^*)$, $P_0$ also needs to have a "positive enough" density function, as otherwise the second term in Eq.(7) would blow up.

**Coverage frequency error** We also note that existing first-order optimization baselines (Bhatnagar et al., 2023; Gibbs & Candès, 2024; Zhang et al., 2024) are equipped with both a regret bound and a coverage frequency error bound, Eq.(4). Hoping to challenge this convention, here we discuss the advantages of only considering the regret.

First, the coverage frequency error is fundamentally "iterate-centric", whereas an ideal performance metric needs to be "data-centric". To be more specific, consider the CP interaction protocol displayed in Figure 1. Achieving low coverage frequency error requires the CP algorithm's output to depend not only on the top level (the nature and the base model), but also on the users' previous confidence level queries. This is in contrast with our regret minimization algorithm, whose output is independent of the users' query history.

Furthermore, just like the pathological example given by Bastani et al. (2022), first-order optimization baselines essentially achieve the desirable coverage frequency due to the "overshooting" provided by the loss linearization. This is perhaps clear from the first online CP algorithm (ACI) proposed by Gibbs & Candès (2021): regarding the update Eq.(3) with the constant learning rate $\eta_t = \eta$, it is shown that the coverage frequency error monotonically decreases as $\eta \to \infty$. Such a peculiar behavior results precisely from overshooting: if $\alpha = 90\%$, then a failed coverage needs nine successful coverages to compensate, and *ensuring* this does not have much to do with the observed data. This casts some natural doubt on the coverage frequency error that the algorithm is designed to optimize.

To reduce the clutter, more discussion on Algorithm 1 is deferred to Section 5. Below we present a few extensions of this core result.

### 3.1 REDUCING MEMORY USAGE VIA QUANTIZATION

Recall our construction of MultiOGD from Section 2. Although not studied by existing works, it is not hard to see that with the size of the grid $\tilde{A}$ being $O(\sqrt{T})$, MultiOGD also satisfies the same $\alpha$-agnostic $O(R\sqrt{T})$ regret bound as in Theorem 2, since the quantile loss $l_\alpha(r, r^*)$ is $R$-Lipschitz with respect to $\alpha$. This raises a natural question: Algorithm 1 requires $O(T)$ memory due to storing the empirical distribution of previous true scores – can we reduce it to $O(\sqrt{T})$?

**Quantized algorithm** Here is a variant of Algorithm 1, denoted as QUANTIZED, achieving this goal. The idea is to discretize the domain $[0, R]$ rather than the $\alpha$-space: we maintain an evenly-spaced grid of size $\sqrt{T}$ over $[0, R]$, round each observed $r_t^*$ to its nearest neighbor $\tilde{r}_t^*$ on the grid, and replace the belief update Eq.(5) by

$$P_t = \lambda_t P_0 + (1 - \lambda_t)\bar{P}(\tilde{r}_{1:t-1}^*).$$

The associated regret bound follows from the Lipschitzness of $l_\alpha(r, r^*)$ with respect to $r^*$.

**Theorem 3.** *With $\lambda_t = 1/\sqrt{t}$ and the uniform $P_0$,* QUANTIZED *achieves* $\text{Regret}_T(\alpha) = O(R\sqrt{T})$.

Compared to MultiOGD, QUANTIZED achieves the same $O(R\sqrt{T})$ regret bound with $O(\sqrt{T})$ memory, while avoiding its monotonicity issue. There is another practical advantage: after observing each $r_t^*$, MultiOGD needs to update all $\sqrt{T}$ base algorithms, whereas QUANTIZED performs only one update on the algorithmic belief $\bar{P}_t$, and then makes $|A_t|$ inferences using the prediction head.

### 3.2 ADAPTIVITY TO IID

In practice, a CP algorithm is often applied without knowing the characteristics of the nature. Previously we have been focusing on the adversarial setting, but what if the true scores $r_{1:T}^*$ turn out to be iid? We now demonstrate the *adaptivity* of Algorithm 1: it automatically achieves almost the same guarantees as ERM under the additional iid assumption.

First, as the coverage probability becomes the default performance metric in the iid setting, we present the following bound on the *dataset-conditional coverage probability*. Notice that the event of successful coverage can be expressed as $r_t^* \le r_t(\alpha)$, where $r_t(\alpha)$ is determined by the past true scores $r_{1:t-1}^*$ and the queried $\alpha$.

**Theorem 4.** *Assume the true score sequence $r_1^*, r_2^*, \ldots$ is drawn iid from an unknown continuous distribution $\mathcal{D}$. With the step size $\lambda_t = 1/\sqrt{t}$ and an arbitrary prior $P_0$, Algorithm 1 guarantees that for any fixed $t \geq 2$, with probability at least $1 - \delta$ over the randomness of $r_{1:t-1}^*$, we have for all $\alpha \in [0, 1]$,*

$$\alpha - \sqrt{\frac{\log(2/\delta)}{2(t-1)}} - \frac{1}{\sqrt{t}-1} \leq \mathbb{P}_{r_t^* \sim \mathcal{D}}\left[r_t^* \leq r_t(\alpha)\right] \leq \alpha + \sqrt{\frac{\log(2/\delta)}{2(t-1)}} + \frac{1}{\sqrt{t}-1} + \frac{1}{t-1}.$$

Compared to the analogous result for ERM (Roth, 2022, Theorem 34), the difference here due to the Bayesian regularization is the $(\sqrt{t}-1)^{-1}$ factor, which is dominated by the existing $O(\sqrt{t^{-1}\log\delta^{-1}})$ term resulting from the randomness. It shows that under the iid assumption, Algorithm 1 achieves almost the same coverage probability error as Split Conformal, despite being designed for the adversarial setting. This is significant as discussed in Section 1.2.

Besides the coverage probability, we can also analyze the *excess quantile risk* of Algorithm 1, which matches the standard oracle inequality one would obtain using ERM.

**Theorem 5.** *Assume the true score sequence $r_1^*, r_2^*, \ldots$ is drawn iid from an unknown distribution $\mathcal{D}$. With the step size $\lambda_t = 1/\sqrt{t}$ and an arbitrary prior $P_0$, Algorithm 1 guarantees that for any fixed $t \geq 2$, with probability at least $1 - \delta$ over the randomness of $r_{1:t-1}^*$, we have for all $\alpha \in [0, 1]$,*

$$\mathbb{E}_{r_t^* \sim \mathcal{D}}[l_\alpha(r_t(\alpha), r_t^*)] \leq \min_{r \in [0,R]} \mathbb{E}_{r_t^* \sim \mathcal{D}}[l_\alpha(r, r_t^*)] + O\left(R\sqrt{\frac{\log(1/\delta)}{t}}\right).$$

### 3.3 CONTINUAL DISTRIBUTION SHIFT

Starting from (Gibbs & Candès, 2021), the study of adversarial online CP has been largely motivated by the prevalence of continual distribution shifts in practice. Tackling this challenge requires *non-converging* algorithms characterized by sliding-window performance guarantees. We now present a discounted variant of Algorithm 1, denoted by DISCOUNTED, along this direction.

**Discounted algorithm** Let $\beta \in (0, 1)$ be a *discount factor*, which is a bandwidth hyperparameter required by DISCOUNTED. Then, we define a regularized and discounted empirical distribution of $r_{1:t}^*$ recursively by

$$\bar{P}_\beta(r_1^*) = \beta P_0 + (1-\beta)\delta(r_1^*),$$

$$\bar{P}_\beta(r_{1:t}^*) = \beta\bar{P}_\beta(r_{1:t-1}^*) + (1-\beta)\delta(r_t^*) = \beta^t P_0 + (1-\beta)\sum_{i=1}^{t}\beta^{t-i}\delta(r_i^*),$$

where $\delta(r_t^*)$ is the distribution with point mass at $r_t^*$. This is used to replace the undiscounted empirical distribution in the belief update, i.e., Eq.(5) is replaced by

$$P_t = \lambda_t P_0 + (1-\lambda_t)\bar{P}_\beta(r_{1:t-1}^*).$$

After that, the prediction head remains unchanged, i.e., $r_t(\alpha) = q_\alpha(P_t)$.

Similar to Theorem 1 and 2, we can show that DISCOUNTED simulates the $\beta$-discounted non-linearized FTRL, which is equipped with a $\beta$-discounted regret bound. Importantly, reasonable step sizes $\lambda_t$ become constant (rather than decreasing), which emphasizes the crucial role of the prior $P_0$: instead of only using $P_0$ to regularize the beginning of the CP game, DISCOUNTED continually mix $P_0$ into its algorithm belief with constant weight, such that it does not "overfit the current environment".

**Theorem 6.** *With $\lambda_t = \lambda = \frac{\sqrt{1-\beta}}{\beta+\sqrt{1-\beta}}$ and the uniform $P_0$, the output $r_t(\alpha)$ of DISCOUNTED satisfies*

$$r_t(\alpha) = \arg\min_{r \in \mathbb{R}}\left[(1-\beta)^{-1}\left(\frac{\lambda}{1-\lambda} + \beta^{t-1}\right)\psi(r) + \sum_{i=1}^{t-1}\beta^{t-1-i}l_\alpha(r, r_i^*)\right],$$

*for all $\alpha$ and $t$. In addition, for all $\alpha \in [0, 1]$, it guarantees the discounted regret bound*

$$\text{Regret}_{T,\beta}(\alpha) := \sum_{t=1}^{T}\beta^{T-t}l_\alpha(r_t(\alpha), r_t^*) - \min_{r \in [0,R]}\sum_{t=1}^{T}\beta^{T-t}l_\alpha(r, r_t^*) \leq \frac{R}{\sqrt{1-\beta}} + o(R),$$

*where $o(\cdot)$ is with respect to $T \to \infty$.*

We remark that (Zhang et al., 2024, Theorem 7) presents a discounted regret lower bound on linear losses, which can be converted to $\Omega(\min\{\alpha, 1-\alpha\}R/\sqrt{1-\beta^2})$ on the quantile losses we consider. Since $(1-\beta)^{-1/2} \leq 2(1-\beta^2)^{-1/2}$ for all $\beta \in (0,1)$, Theorem 6 matches this lower bound in the minimax sense (with respect to $\alpha$, i.e., when $\alpha = 1/2$).

## 4 EXPERIMENT

Complementing our theoretical results, we now evaluate the performance of our Bayesian approach using more experiments. We focus on an actual CP problem: predicting the time-varying volatility of the stock price, with the base model being a standard time series forecasting method called GARCH (Bollerslev, 1986). This experiment was designed by Gibbs & Candès (2021) and further studied by Bastani et al. (2022). See (Bastani et al., 2022, Appendix B.3.1) for the specifics of its context.

Two baselines are considered: a specialization of OGD (ACI) for time series forecasting, and MVP. Besides requiring a fixed learning rate, the former operates on a sliding time window whose length is also a hyperparameter. Similarly, MVP requires picking the size of discretization. For both baselines, we follow the exact implementation from (Bastani et al., 2022), including the hyperparameters.

As for our Bayesian approach, we adopt the discounted version to handle the continual distribution shift, together with quantization. The discretization grid $\tilde{A}$ has the same size as the MVP baseline, and we pick the discount factor $\beta$ such that the effective length $(1-\beta^2)^{-1}$ of the discounted time window exactly matches the length of the ACI baseline's sliding window. Given this $\beta$, $\lambda_t$ is selected according to Theorem 6. It means that compared to the baselines, our algorithm cannot benefit from any extra hyperparameter tuning.

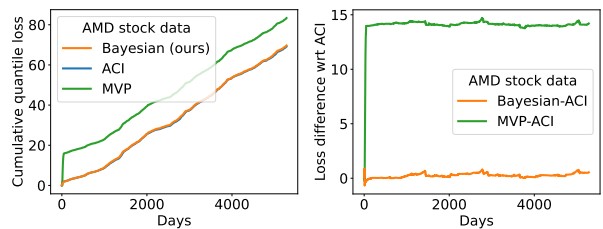

Figure 3: Quantile loss on AMD stock data.

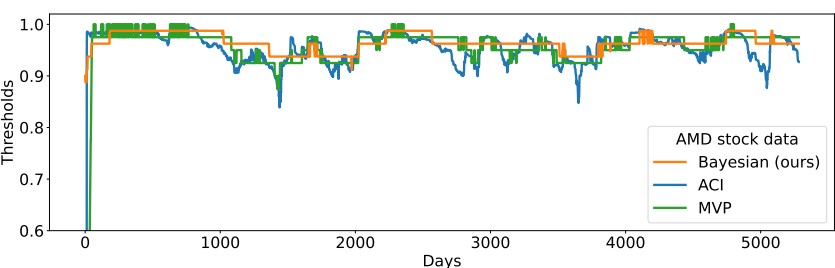

Figure 4: Predicted score threshold on AMD stock data.

With $\alpha = 0.9$, Figure 4 plots the $r_{1:T}(\alpha)$ sequence predicted by different algorithms. As a visual sanity check, our algorithm generates a reasonable prediction sequence with slightly less fluctuation than the baselines. To make a more concrete comparison, Figure 3 plots the total quantile loss suffered by all three algorithms, as well as the difference compared to ACI. It shows that our algorithm achieves almost the same total loss as ACI, and it is faster to warm up than MVP.

Finally, we also evaluate the empirical coverage rate of the tested algorithms. Although our algorithm is not designed for this metric, it performs competitively compared to the baselines. The target is $1 - \alpha = 0.9$, and closer to this target is better. ACI achieves $0.901$, MVP achieves $0.893$, and our Bayesian algorithm achieves $0.899$.

Appendix C includes results on a different stock dataset. It also includes a synthetic experiment where the true sequence switches between 0 and 1; this demonstrates the benefit of our Bayesian algorithm over ERM. Appendix D demonstrates that the monotonicity issue suffered by ACI and MVP also shows up in the stock price experiment (i.e., on real data).

## 5 DISCUSSION

**Loss-agnostic decision making** The downstream simulation of FTRL (Theorem 1) deviates from the common scope of online learning, which requires specifying a single loss function in each round. Instead, it has a similar flavor as the recently proposed concept of *U-calibration* (Kleinberg et al., 2023; Luo et al., 2024): forecasting for an unknown downstream agent. Prior works on U-calibration considered the setting of *finite-class distributional prediction* with generic *proper* losses, while our paper focuses on the continuous domain $[0, R]$ (i.e., "infinitely many classes") with the more specific quantile losses. The extra problem structure allows our algorithm to be deterministic (rather than being randomized like FTPL), thus establishing a closer connection to typical deterministic algorithms in *online convex optimization*.

*Omniprediction* (Gopalan et al., 2022; Garg et al., 2024) is another iconic framework for loss-agnostic decision making, whose main idea is to maintain a *multi-calibrated* algorithmic belief in the sense of (Hébert-Johnson et al., 2018). Our approach does not require calibration as an underlying mechanism.

**Bayesian interpretation** We have been calling our framework "Bayesian", as the belief update Eq.(5) can be viewed by statisticians as a *Bayesian distribution estimator* from iid samples. Following (Gelman et al., 2021, Chapter 23), we now make this very concrete.

Consider the following distribution estimation problem: given $x_1, \ldots, x_n \in \mathcal{X}$ sampled iid from an unknown distribution $X$, what is a good estimate of $X$? As opposed to the frequentist estimate $\bar{P}(x_{1:n})$, a Bayesian estimator would place a prior $F_0$ over all distributions supported on the domain $\mathcal{X}$, compute the posterior $F_n$ from the samples, and output the mean $\mathbb{E}[F_n]$.

For analytical convenience, one would typically choose $F_0$ as a *conjugate prior*: it refers to a family of priors such that if $F_0$ belongs to this family, then $F_n$ also belongs to this family. The most notable conjugate prior for distribution estimation is the *Dirichlet process* (DP), denoted as $\mathrm{DP}(\alpha, P_0)$. Here $\alpha$ and $P_0$ are hyperparameters: $P_0$ equals the mean $\mathbb{E}[\mathrm{DP}(\alpha, P_0)]$, while $\alpha$ controls the variance of $\mathrm{DP}(\alpha, P_0)$. Due to the conjugacy, if $F_0 = \mathrm{DP}(\alpha, P_0)$, then

$$F_n = \mathrm{DP}\left(\alpha + n, \frac{\alpha}{\alpha + n} P_0 + \frac{n}{\alpha + n} \bar{P}(x_{1:n})\right).$$

Consequently, the Bayesian estimator of the distribution $X$ is

$$\mathbb{E}[F_n] = \frac{\alpha}{\alpha + n} P_0 + \frac{n}{\alpha + n} \bar{P}(x_{1:n}).$$

This is the same as the belief update Eq.(5) in our algorithm, with the hyperparameter $\lambda_t = \alpha/(\alpha + n)$. Our results can therefore be regarded as an online adversarial treatment of Bayesian inference, embedded in the CP protocol, and without the iid assumption.

## 6 CONCLUSION

Focusing on the online adversarial formulation of conformal prediction, this paper demonstrates various benefits of being Bayesian. Specifically, we propose a novel Bayesian algorithm with a number of strengths – it supports multiple arbitrary confidence level queries, achieves probably low regret, avoids the monotonicity issue on the obtained confidence sets, and adapts to iid environments. We further develop quantized and discounted extensions of this algorithm, and our theoretical arguments are supported by experiments.

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

## A ADDITIONAL RELATED WORK

This section covers the related work omitted from the main paper. For additional background of CP and its applications, the readers are referred to several excellent resources (Vovk et al., 2005; Roth, 2022; Angelopoulos & Bates, 2023; Tibshirani, 2023).

**Score function** Following (Romano et al., 2020), we first survey a score function for classification, in order to make our setting concrete. Here, given the covariate $x_t$, a ML model would generate a *softmax score* $\pi_t(y)$ for each candidate label $y$, satisfying $\sum_{y \in \mathcal{Y}} \pi_t(y) = 1$. Romano et al. (2020) proposed the score function $s_t(x_t, y) = \sum_{\tilde{y}:\pi_t(\tilde{y}) \geq \pi_t(y)} \pi_t(\tilde{y})$. Consistent with our setting, it is negatively oriented, with the range $R = 1$.

**Bayesian uncertainty quantification** The traditional view of Bayesian methods is statistical: there is a statistical model on which we provide a prior, observe the data, and compute the posterior via the Bayes' theorem. The obtained posterior can then be used to construct confidence sets (called *Bayesian UQ*; Neal 2012), as long as the prior is good enough and the computational procedure is scalable. The key novelty of our work is showing the effectiveness of the Bayesian idea in an adversarial online CP problem. Importantly, no statistical assumptions are imposed on the data, good theoretical performance does not require an unrealistically good prior, and the algorithm also enjoys the computational efficiency of the CP framework. Related but different from our focus, Fong & Holmes (2021) studied a statistical CP problem where the base ML model itself is Bayesian.

## B OMITTED PROOFS

**Theorem 1.** *For all $\alpha \in [0, 1]$, the output $r_t(\alpha)$ of Algorithm 1 satisfies $r_1(\alpha) = \arg\min_{r \in \mathbb{R}} \psi(r)$,*

$$r_t(\alpha) = \arg\min_{r \in \mathbb{R}} \left[ \frac{\lambda_t(t-1)}{1 - \lambda_t} \psi(r) + \sum_{i=1}^{t-1} l_\alpha(r, r_i^*) \right], \quad \forall t \geq 2. \tag{6}$$

*Specifically,*

- *$\psi$ is strongly convex with coefficient $\inf_{r \in [0, R]} p_0(r)$, if the latter is positive.*

- *If $P_0$ is the uniform distribution on $[0, R]$, then $\psi$ is a quadratic function centered at $\alpha R$,*

$$\psi(r) = \frac{1}{2R} r^2 - \alpha r + \frac{1}{2} \alpha R.$$

*Proof of Theorem 1.* We first rewrite the base regularizer $\psi$ as

$$\psi(r) = \int_0^R l_\alpha(r, r^*) p_0(r^*) dr^*$$

$$= (1 - \alpha) \int_0^r (r - r^*) p_0(r^*) dr^* + \alpha \int_r^R (r^* - r) p_0(r^*) dr^*.$$

It is twice-differentiable, with

$$\psi'(r) = (1 - \alpha) \int_0^r p_0(r^*) dr^* - \alpha \int_r^R p_0(r^*) dr^* = \int_0^r p_0(r^*) dr^* - \alpha,$$

and $\psi''(r) = p_0(r)$. The strong convexity statement on $\psi$ is thus clear. If $P_0$ is uniform, we have

$$\psi(r) = R^{-1} \left[ (1 - \alpha) \int_0^r (r - r^*) dr^* + \alpha \int_r^R (r^* - r) dr^* \right]$$

$$= \frac{1}{2R} \left[ (1 - \alpha) r^2 + \alpha (R - r)^2 \right] = \frac{1}{2R} r^2 - \alpha r + \frac{1}{2} \alpha R.$$

Next, consider the first part of the theorem. The case of $t = 1$ is straightforward to verify. For any $t \geq 2$, Algorithm 1 outputs

$$r_t(\alpha) = q_\alpha \left[ \lambda_t P_0 + (1 - \lambda_t) \bar{P}(r^*_{1:t-1}) \right]$$

$$= \min \left\{ r : \lambda_t \int_0^r p_0(r^*) dr^* + \frac{1 - \lambda_t}{t - 1} \sum_{i=1}^{t-1} \mathbf{1}[r^*_i \leq r] \geq \alpha \right\}. \tag{8}$$

On the other hand, consider the optimization objective in Eq.(6), which we write as

$$F_t(r) := \frac{\lambda_t(t - 1)}{1 - \lambda_t} \psi(r) + \sum_{i=1}^{t-1} l_\alpha(r, r^*_i). \tag{9}$$

Notice that the function $F_t(r)$ is continuous and right-differentiable. Taking its right-derivative, we have

$$F'_{t,+}(r) = \frac{\lambda_t(t-1)}{1 - \lambda_t} \left[ \int_0^r p_0(r^*) dr^* - \alpha \right] + \left( -\alpha \sum_{i=1}^{t-1} \mathbf{1}[r < r^*_i] + (1 - \alpha) \sum_{i=1}^{t-1} \mathbf{1}[r \geq r^*_i] \right)$$

$$= \frac{\lambda_t(t-1)}{1 - \lambda_t} \int_0^r p_0(r^*) dr^* - \frac{\alpha \lambda_t(t-1)}{1 - \lambda_t} - \alpha(t-1) + \sum_{i=1}^{t-1} \mathbf{1}[r \geq r^*_i]$$

$$= \frac{t-1}{1 - \lambda_t} \left( \lambda_t \int_0^r p_0(r^*) dr^* + \frac{1 - \lambda_t}{t-1} \sum_{i=1}^{t-1} \mathbf{1}[r \geq r^*_i] - \alpha \right).$$

Comparing it to Eq.(8), we see that the output $r_t(\alpha)$ of Algorithm 1, given by Eq.(8), satisfies

$$r_t(\alpha) = \min\{r : F'_{t,+}(r) \geq 0\}.$$

Since the function $F_t(r)$ is strictly convex, we have $r_t(\alpha) = \arg\min_r F_t(r)$, which is equivalent to Eq.(6). $\qquad \square$

**Theorem 2.** *Let $\mu_{t,\alpha} := \inf\{p_0(r) : r_t(\alpha) \wedge r^*_t \leq r \leq r_t(\alpha) \vee r^*_t\}$. With the step size $\lambda_t = 1/\sqrt{t}$, Algorithm 1 guarantees*

$$\text{Regret}_T(\alpha) := \sum_{t=1}^T l_\alpha(r_t(\alpha), r^*_t) - \sum_{t=1}^T l_\alpha(q_\alpha(r^*_{1:T}), r^*_t) = O\left( \psi(q_\alpha(r^*_{1:T})) \sqrt{T} + \sum_{t=1}^T \frac{1}{\mu_{t,\alpha} \sqrt{t}} \right), \tag{7}$$

*for all time horizon $T$, true score sequence $r^*_{1:T}$, and confidence level $\alpha \in [0, 1]$. Here, $O(\cdot)$ subsumes an absolute constant. Furthermore, if $P_0$ is the uniform distribution on $[0, R]$, then*

$$\text{Regret}_T(\alpha) = O(R\sqrt{T}).$$

*Proof of Theorem 2.* The proof can be decomposed into the following steps.

**Step 1** Starting from the FTRL formulation Eq.(6), we first verify that the regularizer weight $\frac{\lambda_t(t-1)}{1 - \lambda_t}$ is increasing with respect to $t$ (when $t > 1$), which is required by the FTRL analysis. To this end, define

$$h_t := \frac{\lambda_t(t-1)}{1 - \lambda_t} = \frac{t - 1}{\sqrt{t} - 1}.$$

Taking the derivative with respect to $t$, for all $t > 1$,

$$\frac{dh_t}{dt} = \frac{\sqrt{t} - 1 - \frac{t-1}{2\sqrt{t}}}{(\sqrt{t} - 1)^2} = \frac{t - 2\sqrt{t} + 1}{2\sqrt{t}(\sqrt{t+1} - 1)^2} = \frac{(\sqrt{t} - 1)^2}{2\sqrt{t}(\sqrt{t+1} - 1)^2} \geq 0.$$

For completeness, we also define $h_1 = 1$.

Besides, we have the order estimate $h_t = O(\sqrt{t})$, $1/h_t = O(1/\sqrt{t})$, where $O(\cdot)$ only hides an absolute constant.

**Step 2**  Next, due to Theorem 1, we can apply the standard FTRL analysis. Recall our notation from Eq.(9): we write the optimization objective in Eq.(6) as

$$F_t(r) := h_t \psi(r) + \sum_{i=1}^{t-1} l_\alpha(r, r_i^*), \quad \forall t \geq 2.$$

Similarly, we also write $F_1(r) := h_1 \psi(r)$. Notice that $r_t(\alpha) = \arg\min_{r \in \mathbb{R}} F_t(r)$ for all $t$.

The classical FTRL equality lemma (Orabona, 2023, Lemma 7.1) states that

$$\text{Regret}_T(\alpha) = h_{T+1} \psi(q_\alpha(r_{1:T}^*)) - \min_{r \in \mathbb{R}} \psi(r) + \sum_{t=1}^{T} [F_t(r_t(\alpha)) - F_{t+1}(r_{t+1}(\alpha)) + l_\alpha(r_t(\alpha), r_t^*)]$$

$$+ F_{T+1}(r_{T+1}(\alpha)) - F_{T+1}(q_\alpha(r_{1:T}^*))$$

$$\leq h_{T+1} \psi(q_\alpha(r_{1:T}^*)) + \sum_{t=1}^{T} [F_t(r_t(\alpha)) - F_{t+1}(r_{t+1}(\alpha)) + l_\alpha(r_t(\alpha), r_t^*)],$$

where the second line is due to $\min_r \psi(r) \geq 0$, and $r_{T+1}(\alpha) = \arg\min_{r \in \mathbb{R}} F_{T+1}(r)$.

Consider the sum on the RHS, where for conciseness we omit $(\alpha)$ in the notation. This is the typical one-step quantity involved in the FTRL analysis. Following a similar procedure as (Orabona, 2023, Lemma 7.8), we have

$$F_t(r_t) - F_{t+1}(r_{t+1}) + l_\alpha(r_t, r_t^*)$$
$$= F_t(r_t) + l_\alpha(r_t, r_t^*) - F_t(r_{t+1}) - l_\alpha(r_{t+1}, r_t^*) + (h_t - h_{t+1})\psi(r_{t+1})$$
$$\leq F_t(r_t) + l_\alpha(r_t, r_t^*) - F_t(r_{t+1}) - l_\alpha(r_{t+1}, r_t^*) \qquad (h_{t+1} \geq h_t, \psi(r_{t+1}) \geq 0)$$
$$\leq F_t(r_t) + l_\alpha(r_t, r_t^*) - \min_{r \in \mathbb{R}} [F_t(r) + l_\alpha(r, r_t^*)].$$

Observe that since $F_t(\cdot)$ and $l_\alpha(\cdot, r_t^*)$ are both convex, the minimizing argument of their sum lies between their respective unique minimizers, $r_t$ and $r_t^*$. On this segment, the function $F_t$ is $h_t\mu_{t,\alpha}$-strongly-convex, where $\mu_{t,\alpha}$ is defined in the assumption of the theorem. We now proceed using the property of strong convexity (Orabona, 2023, Lemma 7.6), which we restate as Lemma B.1.

Concretely, if $g_t$ is a subgradient of $l_\alpha(\cdot, r_t^*)$ at $r_t$, then it is also a subgradient of $F_t(\cdot) + l_\alpha(\cdot, r_t^*)$ at $r_t$, since $r_t = \arg\min_r F_t(r)$. Moreover, such a subgradient $g_t$ satisfies $|g_t| \leq 1$ due to $l_\alpha(\cdot, r_t^*)$ being 1-Lipschitz. Combining these with the strong convexity, Lemma B.1 yields

$$F_t(r_t) + l_\alpha(r_t, r_t^*) - \min_{r \in \mathbb{R}} [F_t(r) + l_\alpha(r, r_t^*)] \leq \frac{1}{2h_t\mu_{t,\alpha}}.$$

Plugging this all the way back into the regret bound, we have

$$\text{Regret}_T(\alpha) \leq h_{T+1} \psi(q_\alpha(r_{1:T}^*)) + \frac{1}{2} \sum_{t=1}^{T} \frac{1}{h_t\mu_{t,\alpha}}$$

$$= O\left( \psi(q_\alpha(r_{1:T}^*))\sqrt{T} + \sum_{t=1}^{T} \frac{1}{\mu_{t,\alpha}\sqrt{t}} \right).$$

**Step 3**  Finally we analyze the special case of uniform prior. From Theorem 1,

$$\psi(q_\alpha(r_{1:T}^*)) \leq \max_{r \in [0,R]} \psi(r) \leq \max_{r \in [0,R]} \left( \frac{1}{2R}r^2 - \alpha r + \frac{1}{2}\alpha R \right) \leq \frac{R}{2}.$$

Furthermore, $\mu_{t,\alpha} = 1/R$. Plugging in $\sum_{t=1}^{T} t^{-1/2} = O(\sqrt{T})$ completes the proof. □

**Theorem 3.** *With $\lambda_t = 1/\sqrt{t}$ and the uniform $P_0$,* QUANTIZED *achieves* $\text{Regret}_T(\alpha) = O(R\sqrt{T})$.

*Proof of Theorem 3.* Recall from Section 3.1 that the quantized true score is denoted by $\tilde{r}_t^*$. From Theorem 2, we have

$$\sum_{t=1}^{T} l_\alpha(r_t(\alpha), \tilde{r}_t^*) - \sum_{t=1}^{T} l_\alpha(q_\alpha(r_{1:T}^*), \tilde{r}_t^*) \leq \sum_{t=1}^{T} l_\alpha(r_t(\alpha), \tilde{r}_t^*) - \sum_{t=1}^{T} l_\alpha(q_\alpha(\tilde{r}_{1:T}^*), \tilde{r}_t^*)$$

$$= O(R\sqrt{T}).$$

As $|\tilde{r}_t^* - r_t^*| \leq R/\sqrt{T}$ and the quantile loss function $l_\alpha(r, r^*)$ is 1-Lipschitz with respect to $r^*$, we have

$$\left| \sum_{t=1}^{T} l_\alpha(r_t(\alpha), \tilde{r}_t^*) - \sum_{t=1}^{T} l_\alpha(r_t(\alpha), r_t^*) \right| \leq \sum_{t=1}^{T} |l_\alpha(r_t(\alpha), \tilde{r}_t^*) - l_\alpha(r_t(\alpha), r_t^*)|$$

$$\leq \sum_{t=1}^{T} |\tilde{r}_t^* - r_t^*| \leq R\sqrt{T}.$$

The comparator term $\sum_{t=1}^{T} l_\alpha(q_\alpha(r_{1:T}^*), \tilde{r}_t^*)$ can be related similarly to $\sum_{t=1}^{T} l_\alpha(q_\alpha(r_{1:T}^*), r_t^*)$, and combining the above completes the proof. □

**Theorem 4.** *Assume the true score sequence $r_1^*, r_2^*, \ldots$ is drawn iid from an unknown continuous distribution $\mathcal{D}$. With the step size $\lambda_t = 1/\sqrt{t}$ and an arbitrary prior $P_0$, Algorithm 1 guarantees that for any fixed $t \geq 2$, with probability at least $1 - \delta$ over the randomness of $r_{1:t-1}^*$, we have for all $\alpha \in [0, 1]$,*

$$\alpha - \sqrt{\frac{\log(2/\delta)}{2(t-1)}} - \frac{1}{\sqrt{t}-1} \leq \mathbb{P}_{r_t^* \sim \mathcal{D}} [r_t^* \leq r_t(\alpha)] \leq \alpha + \sqrt{\frac{\log(2/\delta)}{2(t-1)}} + \frac{1}{\sqrt{t}-1} + \frac{1}{t-1}.$$

*Proof of Theorem 4.* The proof follows a similar strategy as (Roth, 2022, Theorem 34). First, for any fixed $t \geq 2$, the samples $r_{1:t-1}^*$ have no ties almost surely, since the underlying distribution $\mathcal{D}$ is continuous. We will condition the rest of the analysis on this event.

Next, recall Algorithm 1's prediction rule, Eq.(8). On one hand, we have

$$\lambda_t \int_0^{r_t(\alpha)} p_0(r^*) dr^* + \frac{1 - \lambda_t}{t - 1} \sum_{i=1}^{t-1} \mathbf{1}[r_i^* \leq r_t(\alpha)] \geq \alpha,$$

which means

$$\frac{1 - \lambda_t}{t - 1} \sum_{i=1}^{t-1} \mathbf{1}[r_i^* \leq r_t(\alpha)] \geq \alpha - \lambda_t,$$

$$\frac{1}{t - 1} \sum_{i=1}^{t-1} \mathbf{1}[r_i^* \leq r_t(\alpha)] \geq \alpha + \frac{\lambda_t}{1 - \lambda_t}(\alpha - 1) \geq \alpha - \frac{1}{\sqrt{t}-1}.$$

On the other hand, if we define $m = \sum_{i=1}^{t-1} \mathbf{1}[r_i^* \leq r_t(\alpha)]$ and let $r_{-1}^*$ be the $(m-1)$-th smallest element of $r_{1:t-1}^*$, then it is also clear from Eq.(8) that

$$\lambda_t \int_0^{r_{-1}^*} p_0(r^*) dr^* + \frac{1 - \lambda_t}{t - 1} \sum_{i=1}^{t-1} \mathbf{1}[r_i^* \leq r_{-1}^*] \leq \alpha,$$

which means

$$\frac{1 - \lambda_t}{t - 1} \sum_{i=1}^{t-1} \mathbf{1}[r_i^* \leq r_{-1}^*] \leq \alpha - \lambda_t \int_0^{r_{-1}^*} p_0(r^*) dr^* \leq \alpha,$$

$$\frac{1}{t - 1} \sum_{i=1}^{t-1} \mathbf{1}[r_i^* \leq r_{-1}^*] \leq \frac{\alpha}{1 - \lambda_t} \leq \alpha + \frac{1}{\sqrt{t}-1},$$

$$\frac{1}{t - 1} \sum_{i=1}^{t-1} \mathbf{1}[r_i^* \leq r_t(\alpha)] \leq \frac{1}{t - 1} \sum_{i=1}^{t-1} \mathbf{1}[r_i^* \leq r_{-1}^*] + \frac{1}{t - 1} \leq \alpha + \frac{1}{\sqrt{t}-1} + \frac{1}{t-1}.$$

In summary,

$$\alpha - \frac{1}{\sqrt{t}-1} \leq \frac{1}{t - 1} \sum_{i=1}^{t-1} \mathbf{1}[r_i^* \leq r_t(\alpha)] \leq \alpha + \frac{1}{\sqrt{t}-1} + \frac{1}{t-1}. \tag{10}$$

Finally we apply the DKW inequality (Lemma B.3). For all $\varepsilon > 0$, we have

$$\mathbb{P}_{r_{1:t-1}^*}\left[\sup_{\alpha \in [0,1]}\left|\left(\frac{1}{t-1}\sum_{i=1}^{t-1}\mathbf{1}[r_i^* \leq r_t(\alpha)]\right) - \mathbb{P}_{r_t^*}[r_t^* \leq r_t(\alpha)]\right| > \varepsilon\right] \leq 2\exp\left[-2(t-1)\varepsilon^2\right].$$

Therefore, with probability at least $1 - \delta$ over the randomness of $r_{1:t-1}^*$, we have

$$\left|\left(\frac{1}{t-1}\sum_{i=1}^{t-1}\mathbf{1}[r_i^* \leq r_t(\alpha)]\right) - \mathbb{P}_{r_t^*}[r_t^* \leq r_t(\alpha)]\right| \leq \sqrt{\frac{\log(2/\delta)}{2(t-1)}}, \quad \forall \alpha \in [0,1].$$

Combining it with Eq.(10) above completes the proof. $\qquad\square$

**Theorem 5.** *Assume the true score sequence $r_1^*, r_2^*, \ldots$ is drawn iid from an unknown distribution $\mathcal{D}$. With the step size $\lambda_t = 1/\sqrt{t}$ and an arbitrary prior $P_0$, Algorithm 1 guarantees that for any fixed $t \geq 2$, with probability at least $1 - \delta$ over the randomness of $r_{1:t-1}^*$, we have for all $\alpha \in [0,1]$,*

$$\mathbb{E}_{r_t^* \sim \mathcal{D}}[l_\alpha(r_t(\alpha), r_t^*)] \leq \min_{r \in [0,R]}\mathbb{E}_{r_t^* \sim \mathcal{D}}[l_\alpha(r, r_t^*)] + O\left(R\sqrt{\frac{\log(1/\delta)}{t}}\right).$$

*Proof of Theorem 5.* The proof follows from a standard uniform convergence argument (Zhang, 2023) combined with the Lipschitzness of the quantile loss.

First, notice that with any combination of $\alpha$, $r$ and $r^*$, the quantile loss $l_\alpha(r, r^*) \in [0, R]$. Therefore, fixing any $\alpha \in [0,1]$ and $r \in [0,R]$, we apply the Hoeffding's inequality (Lemma B.2) to obtain

$$\mathbb{P}_{r_{1:t-1}^*}\left[\left|\frac{1}{t-1}\sum_{i=1}^{t-1}l_\alpha(r, r_i^*) - \mathbb{E}_{r_t^*}[l_\alpha(r, r_t^*)]\right| \geq \varepsilon\right] \leq 2\exp\left(-\frac{2(t-1)\varepsilon^2}{R^2}\right).$$

Next, we evenly discretize $[0,1]$ by a grid of size $\sqrt{t}$, and also $[0,R]$ by a grid of size $\sqrt{t}$, and denote their combination as a set $S$. $|S| = t$. For all $\alpha$ and $r$, there exists $(\tilde{\alpha}, \tilde{r}) \in S$ satisfying $|\alpha - \tilde{\alpha}| \leq 1/\sqrt{t}$ and $|r - \tilde{r}| \leq R/\sqrt{t}$. Applying the union bound on $S$ yields

$$\mathbb{P}_{r_{1:t-1}^*}\left[\max_{(\alpha,r) \in S}\left|\frac{1}{t-1}\sum_{i=1}^{t-1}l_\alpha(r, r_i^*) - \mathbb{E}_{r_t^*}[l_\alpha(r, r_t^*)]\right| \geq \varepsilon\right] \leq 2t\exp\left(-\frac{2(t-1)\varepsilon^2}{R^2}\right),$$

which means with probability at least $1 - \delta$,

$$\max_{(\alpha,r) \in S}\left|\frac{1}{t-1}\sum_{i=1}^{t-1}l_\alpha(r, r_i^*) - \mathbb{E}_{r_t^*}[l_\alpha(r, r_t^*)]\right| \leq R\sqrt{\frac{\log(2t/\delta)}{2(t-1)}}.$$

Since $l_\alpha(r, r^*)$ is $R$-Lipschitz with respect to $\alpha$, and 1-Lipschitz with respect to $r$, we have

$$\max_{0 \leq \alpha \leq 1, 0 \leq r \leq R}\left|\frac{1}{t-1}\sum_{i=1}^{t-1}l_\alpha(r, r_i^*) - \mathbb{E}_{r_t^*}[l_\alpha(r, r_t^*)]\right| \leq R\sqrt{\frac{\log(2t/\delta)}{2(t-1)}} + \frac{2R}{\sqrt{t}}.$$

Finally, due to Theorem 1 we have for all $\alpha$ and $r$,

$$\frac{1}{t-1}\sum_{i=1}^{t-1}l_\alpha(r_t(\alpha), r_i^*) \leq \frac{1}{t-1}\sum_{i=1}^{t-1}l_\alpha(r, r_i^*) + \frac{\lambda_t}{1-\lambda_t}\left[\psi(r) - \psi(r_t(\alpha))\right]$$

$$\leq \frac{1}{t-1}\sum_{i=1}^{t-1}l_\alpha(r, r_i^*) + \frac{1}{\sqrt{t}-1}\max\left\{\psi(0), \psi(R)\right\}$$

$$\leq \frac{1}{t-1}\sum_{i=1}^{t-1}l_\alpha(r, r_i^*) + \frac{R}{\sqrt{t}-1}.$$

Combining it with the generalization error bound above, with high probability we have for all $\alpha$ and $r$,

$$\mathbb{E}_{r_t^*}[l_\alpha(r_t(\alpha), r_t^*)] \leq \mathbb{E}_{r_t^*}[l_\alpha(r, r_t^*)] + \frac{R}{\sqrt{t}-1} + 2\left(R\sqrt{\frac{\log(2t/\delta)}{2(t-1)}} + \frac{2R}{\sqrt{t}}\right).$$

Taking $\min_r$ on the RHS completes the proof. $\qquad\square$

**Theorem 6.** *With $\lambda_t = \lambda = \frac{\sqrt{1-\beta}}{\beta + \sqrt{1-\beta}}$ and the uniform $P_0$, the output $r_t(\alpha)$ of* DISCOUNTED *satisfies*

$$r_t(\alpha) = \arg\min_{r \in \mathbb{R}} \left[ (1-\beta)^{-1} \left( \frac{\lambda}{1-\lambda} + \beta^{t-1} \right) \psi(r) + \sum_{i=1}^{t-1} \beta^{t-1-i} l_\alpha(r, r_i^*) \right],$$

*for all $\alpha$ and $t$. In addition, for all $\alpha \in [0, 1]$, it guarantees the discounted regret bound*

$$\text{Regret}_{T,\beta}(\alpha) := \sum_{t=1}^{T} \beta^{T-t} l_\alpha(r_t(\alpha), r_t^*) - \min_{r \in [0,R]} \sum_{t=1}^{T} \beta^{T-t} l_\alpha(r, r_t^*) \le \frac{R}{\sqrt{1-\beta}} + o(R),$$

*where $o(\cdot)$ is with respect to $T \to \infty$.*

*Proof of Theorem 6.* Analogous to Eq.(8), we can write the output of DISCOUNTED as

$$r_t(\alpha) = q_\alpha \left[ \lambda P_0 + (1-\lambda) \bar{P}_\beta(r_{1:t-1}^*) \right]$$

$$= \min \left\{ r : \frac{r}{R} \left[ \lambda + \beta^{t-1}(1-\lambda) \right] + (1-\lambda)(1-\beta) \sum_{i=1}^{t-1} \beta^{t-1-i} \mathbf{1}[r_i^* \le r] \ge \alpha \right\}.$$

Similar to Eq.(9), this can be verified as a minimizer of the objective

$$H_t(r) := (1-\beta)^{-1} \left( \frac{\lambda \beta^{1-t}}{1-\lambda} + 1 \right) \psi(r) + \sum_{i=1}^{t-1} \beta^{-i} l_\alpha(r, r_i^*).$$

For the convenience of notation, we will write the regularizer weight as $z_t := (1-\beta)^{-1} \left( \frac{\lambda \beta^{1-t}}{1-\lambda} + 1 \right)$.

Notice that with the uniform $P_0$, the base regularizer $\psi$ is $R^{-1}$-strongly-convex due to Theorem 1, therefore we can apply the strong-convexity-based FTRL analysis (Orabona, 2023, Corollary 7.9) on the scaled loss functions,

$$h_t(r) := \beta^{-t} l_\alpha(r, r_t^*).$$

This yields

$$\sum_{t=1}^{T} h_t(r_t(\alpha)) - \min_{r \in [0,R]} \sum_{t=1}^{T} h_t(r) \le z_T \left[ \max_{r \in [0,R]} \psi(r) - \min_{r \in [0,R]} \psi(r) \right] + \frac{R}{2} \sum_{t=1}^{T} \frac{g_t^2}{z_t^2},$$

where $g_t$ can be any subgradient of $h_t(r)$ at $r = r_t(\alpha)$. Scaling both sides by $\beta^T$, we recover the discounted regret definition on the LHS:

$$\text{Regret}_{T,\beta}(\alpha) \le \beta^T z_T \left[ \max_{r \in [0,R]} \psi(r) - \min_{r \in [0,R]} \psi(r) \right] + \frac{R\beta^T}{2} \sum_{t=1}^{T} \frac{g_t^2}{z_t}.$$

Next we simplify the obtained expression. The range of $\phi$ is contained in $[0, R/2]$. In addition, $|g_t| \le \beta^{-t}$ since the quantile loss $l_\alpha(r, r^*)$ is 1-Lipschitz with respect to $r$. Therefore,

$$\sum_{t=1}^{T} \frac{g_t^2}{z_t} \le \sum_{t=1}^{T} \frac{\beta^{-2t}}{(1-\beta)^{-1} \left( \frac{\lambda \beta^{1-t}}{1-\lambda} + 1 \right)} \le \frac{1-\beta}{\beta} \frac{1-\lambda}{\lambda} \sum_{t=1}^{T} \beta^{-t} \le \frac{1-\lambda}{\lambda} \beta^{-T-1},$$

$$\text{Regret}_{T,\beta}(\alpha) \le \frac{R}{2} \left( \frac{\beta^T}{1-\beta} + \frac{\lambda}{1-\lambda} \frac{\beta}{1-\beta} + \frac{1-\lambda}{\lambda} \frac{1}{\beta} \right).$$

Notice that our choice of $\lambda$ satisfies $\frac{\lambda}{1-\lambda} = \beta^{-1}\sqrt{1-\beta}$, therefore

$$\text{Regret}_{T,\beta}(\alpha) \le \frac{R}{2} \left( \frac{\beta^T}{1-\beta} + \frac{2}{\sqrt{1-\beta}} \right) = \frac{R}{\sqrt{1-\beta}} + o(R),$$

where $o(\cdot)$ is with respect to $T \to \infty$. $\qquad\square$

### B.1 AUXILIARY LEMMA

**Lemma B.1** (Lemma 7.6 of (Orabona, 2023)). *Let $f$ be a $\mu$-strongly convex function with respect to a norm $\|\cdot\|$, over a convex set $V$. For all $x, y \in V$ and subgradients $g \in \partial f(y)$, $g' \in \partial f(x)$, we have*

$$f(x) - f(y) \leq \langle g, x - y \rangle + \frac{1}{2\mu} \|g - g'\|_*^2.$$

*Here $\langle \cdot, \cdot \rangle$ denotes the inner product, and $\|\cdot\|_*$ denotes the dual norm of $\|\cdot\|$.*

The following lemma is a standard tool in ML due to (Hoeffding, 1963).

**Lemma B.2** (Hoeffding's inequality). *Let $x_1, \ldots, x_n$ be iid samples of a real-valued random variable on $[a, b]$. Let $\bar{x}$ be the mean of the distribution. Then, for all $\varepsilon > 0$, we have*

$$\mathbb{P}\left[\left|\frac{1}{n}\sum_{i=1}^{n} x_i - \bar{x}\right| \geq \varepsilon\right] \leq 2\exp\left(-\frac{2n\varepsilon^2}{(b-a)^2}\right).$$

The next lemma is the celebrated Dvoretzky–Kiefer–Wolfowitz inequality, due to (Dvoretzky et al., 1956; Massart, 1990).

**Lemma B.3** (DKW inequality). *Let $x_1, \ldots, x_n$ be iid samples of a real-valued random variable with cumulative distribution function $F$, and let $\bar{P}(x_{1:n})$ be the empirical distribution of $x_{1:n}$, with cumulative distribution function $\hat{F}_n$. For all $\varepsilon > 0$, we have*

$$\mathbb{P}\left[\sup_{x \in \mathbb{R}}\left|\hat{F}_n(x) - F(x)\right| > \varepsilon\right] \leq 2\exp(-2n\varepsilon^2).$$

## C ADDITIONAL EXPERIMENT

Extending Section 4, this section presents the result of our stock price experiment using a different dataset (NVDA instead of AMD). The same procedure from Section 4 is followed. Figure 5 plots the predicted thresholds, and Figure 6 plots the total quantile loss. Overall they exhibit the similar behavior as the result from Section 4.

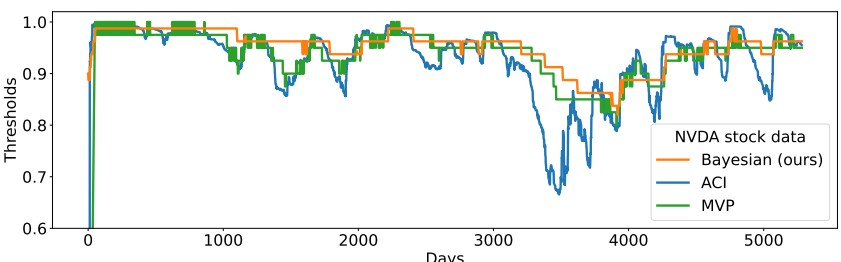

Figure 5: Predicted score threshold on NVDA stock data.

As for the coverage frequency, ACI achieves $0.899$, MVP achieves $0.891$, and our Bayesian algorithm achieves $0.897$. Again, closer to the target $0.9$ is better. The conclusion is that in the fixed-$\alpha$ setting our algorithm performs competitively compared to the baselines, while in the multi-$\alpha$ setting it demonstrates the advantage from Section 2.

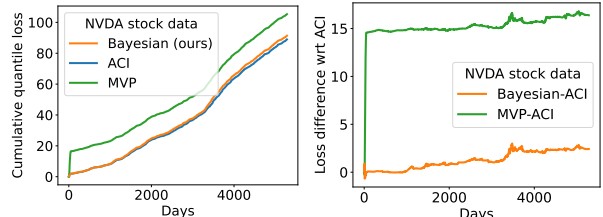

Figure 6: Quantile loss on NVDA stock data.

**Switching sequence** Besides, to demonstrate the failure of ERM without the iid assumption, we consider a synthetic $r_{1:T}^*$ sequence which switches in every round between 0

and 1. Similar to Section 2, four algorithms are tested: OGD (Gibbs & Candès, 2021), MVP (Bastani et al., 2022),[3] ERM and our Bayesian algorithm QUANTIZED. Figure 7 plots their regret measured by the quantile loss, under two different $\alpha$ values.

Consistent with the classical online learning theory, ERM becomes brittle when $\alpha$ matches the long run average of $r^*_{1:T}$ (i.e, 0.5), suffering linear regret with respect to $T$. In contrast, both OGD (with $\eta_t = t^{-1/2}$; $\alpha$ is known) and our Bayesian algorithm achieve sublinear regret under both $\alpha$ values. Quite different from the conventional online learning framework, MVP is designed to minimize the conditional empirical coverage error, but nonetheless, it achieves low regret when $\alpha = 0.5$. The limitation is that MVP requires a relatively long period to warm up: when $\alpha = 0.7$, the regret of MVP grows linearly at the beginning, before hitting a plateau at $T \approx 800$.

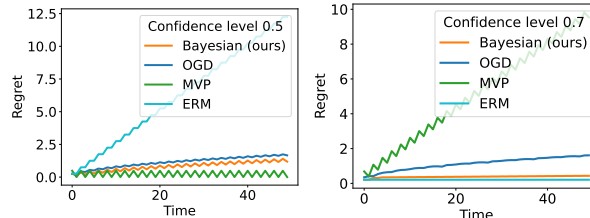

Figure 7: Regret on switching data.

[3]Similar to Section 2, we use a single random seed for the MVP baseline throughout this section, since we find the results to be generally insensitive to the seed.

## D    ADDITIONAL RESULT FOR REBUTTAL

In this section we present more results to address the reviewers' comments.

First, we extend our previous experiment on stock data to show that the monotonicity issue demonstrated in Section 2 actually also shows up in this concrete application. We adopt the same hyperparameters as the other stock experiments, which means that our algorithm does not benefit from any extra hyperparameter tuning. The AMD dataset is considered.

For all three algorithm (ACI, MVP, DISCOUNTED), we run two copies with different $\alpha$ values: $\alpha_1 = 0.9$ and $\alpha_2 = 0.905$. For MVP which is randomized, this also means that the two copies are given exactly the same random number sequence. In principle, the predicted score threshold corresponding to $\alpha_2$ is larger than the threshold corresponding to $\alpha_1$. We plot their difference sequences in Figure 8; if there is a negative value, then it means the monotonicity in the sense of Section 2 is violated. The result shows that our Bayesian algorithm satisfies such monotonicity while ACI and MVP do not, which demonstrates its practical advantage when multiple confidence levels are of interest.

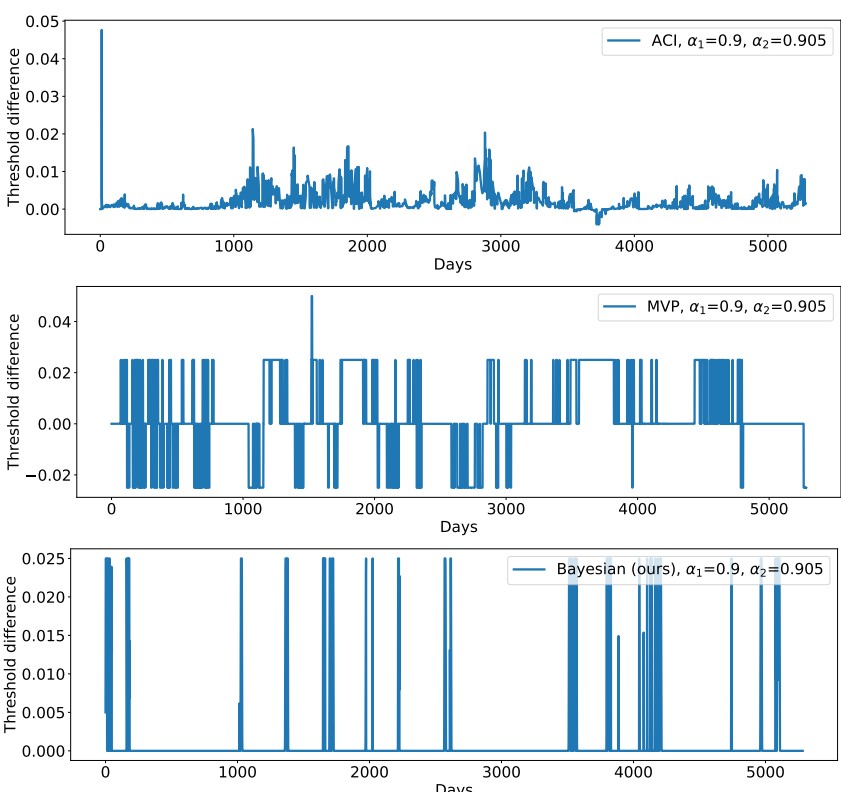

Figure 8: The difference between the two predicted threshold sequences corresponding to $\alpha_1 = 0.9$ and $\alpha_2 = 0.905$. The monotonicity property is violated if the plotted sequence has negative value. The results show that our Bayesian algorithm satisfies such monotonicity whereas ACI and MVP do not.

Next, responding to Reviewer e4ek, we would like to show that our discounted algorithm can indeed handle distribution shifts (which is consistent with the discounted regret bound, Theorem 6). From the latest reply by Reviewer e4ek, it appears that such a concern has been dismissed. But since we have already performed the experiment, we thought it would still be helpful to include the results for the readers' information.

To this end, we consider the experimental setting suggested by Reviewer e4ek, which is also considered in (Bastani et al., 2022): let the true score sequence be monotonically increasing. We use exactly

the same setting as (Bastani et al., 2022), which means that all three algorithms we test (ACI, MVP and DISCOUNTED) as well as their hyperparameters are still the same as our stock price experiments.

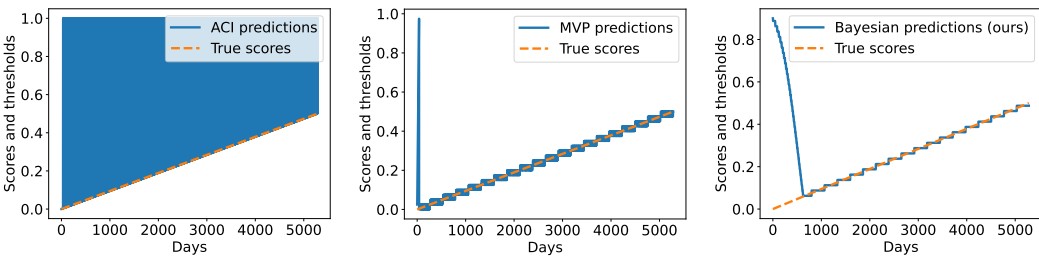

Figure 9: Predicted score thresholds on a synthetic sequence of linearly increasing true scores.

The results are shown in Figure 9. The orange dashed line shows a synthetic, linearly growing sequence, which is used as the true score sequence $r^*_{1:T}$. The blue solid line plots the predicted score thresholds of each algorithm. It can be seen that the predictions of both ACI and our algorithm (DISCOUNTED) can faithfully track the growing trend, whereas ACI suffers from an undesirable oscillatory behavior which is also discussed in (Bastani et al., 2022) (essentially, the predicted threshold sequence of ACI alternates between just below the true score and the maximum value which is 1).

