# OpenReview forum: "The Benefit of Being Bayesian in Online Conformal Prediction"
_ICLR.cc/2025/Conference — Submitted to ICLR 2025_

### Official Review · Reviewer_TgXn · 2024-10-31

**Soundness:** 4
**Presentation:** 4
**Contribution:** 3
**Rating:** 8
**Confidence:** 2

**Summary:**

The authors propose a Bayesian frame work for comformal prediction, bridging the two scenarios of "data-centric" approach where usually iid/exchangibility assumption is required, and the "iterate-centric" approach which is more robust towards distributional shift.

The Bayescian CP method has a few desired properties: the confidence sets are montone in the nominal confidence level $\alpha$ for the online prediction; the $O(\sqrt{T})$ regret bound is achieved; the method adapts to both the iid case and distributional shift case with properly chosen step size.

**Strengths:**

The paper is well-written and clear to its points. The Bayesian method proposed is easy to understand and implement. The theoretical guarantees of a tight regret bound is provided, and the CIs are monotone in the nominal level $\alpha$. The method enjoys both desired properties for "data-centric" and "iterate-centric" approaches, and is able to recover previous known bounds under properly chosen learning rate. Numerical experiments comparing to previous benchmarks were provided.

**Weaknesses:**

1. The uniform prior is robust, but as the authors had mentioned, if certain knowledge is known a tighter bound could be achieved potentially. I would be interested to see more fine-grained analysis in this scenario, and whether there is a way to update the regularized belief instead of using the same one upon observing more data.
2. For the continual distribution shift case, the learning rate is constant for the discounted regret. There seems to be a gap of choosing diminising ($O(1/\sqrt{t})$) step size and $O(1)$ step size, and how to choose this adaptively.

**Questions:**

1. Is it possible to achieve lower regret by adpatively choosing the regularized belief?
2. Is there a way to detect distributional shift such that the step size can be chosen adaptively? Is there a uniform way to measure the performance/regret for both the iid/exchangeable case and the distribution shift case?

---

> ### Author Response · Authors · 2024-11-14
>
> Thank you very much for your review!
>
> Updating the prior online is indeed an interesting idea worth looking into in the future. As for unifying the regimes of decaying step sizes and constant step sizes, one might consider characterizing the *dynamic regret* (Zinkevich, 2003), which is another common performance metric in online learning. Algorithms targeting this performance metric usually have a "meta-learning" type of structure, where a number of "base algorithms" (like our Algorithm 1) with different learning rates are aggregated by a meta-algorithm on top (which in some sense "selects" the best learning rate). It's quite likely that extending our approach towards this direction would yield new theoretical results, but that would typically sacrifice the practicality (while not necessarily leading to new insights), which is why we did't pursue this direction in the present work.

---

### Official Review · Reviewer_e4ek · 2024-11-03

**Soundness:** 3
**Presentation:** 1
**Contribution:** 2
**Rating:** 5
**Confidence:** 4

**Summary:**

The paper proposes a method for online conformal prediction leveraging a Follow-the-Regularized-Leader approach, where the regularization is derived from a prior. Theoretical justification for the method is provided by the means of regret bounds. The method is empirically evaluated on synthetic data as well as financial time series.

**Strengths:**

The key idea of having an online conformal prediction method mimic online bayesian inference methods is interesting, and, as far as I am aware, original. The text is reasonably clear, and the authors provide a substantial number of theoretical guarantees, substantiating their method's strength. The inclusion of analyses in the IID case (i.e., not just an arbitrarily adversarial case) is appreciated.

**Weaknesses:**

The paper has a number of significant weaknesses.

- Line 117: The first paradox does not seem paradoxical at all. We are in a setting where, in principle, exchangeability does not hold. That precisely means that the order in which we have observed our data carries information. So our method *should not* be invariant to this order!
- The paper is about online conformal prediction, and thus for settings where exchangeability & IID do not hold. However, the paper focuses excessively on the IID setting, except for a single section (Section 3.3), which introduces a separate algorithm (dubbed 'Discounted') and a corresponding theorem (Theorem 6), and Theorem 2. And Theorem 6 does not seem to agree with the text surrounding it; the 'optimal' predictions considered for the regret bound seems to be constant over the course of the whole time (since the min over $r$ is outside the sum over time), which is deeply uninteresting. Any minimally decent predictor in the continual distribution shift setting should have predictions that change over time. Theorem 2 suffers from the same issue. (A simple case where these things really matter is when your are in an adversarial setting with increasing conformity scores over time. No single prediction is going to be enough, you must increase your intervals over time!)
- The experimental evaluation is very weak. Online conformal prediction is a topic that requires significant experimental evaluation. The paper's theoretical contributions aren't notable enough nor are any new things possible with the proposed method, so an advantage of the method validated through experiments (e.g., that the coverage would be more robust, or that the intervals would be smaller) is essential.
    - The only data considered was one synthetic dataset and stock market data. The main problem is, of course, with the real data; only stock market data was considered. Data coming from the stock market (and especially price time series) have very particular structure. The authors should consider many other time series. Moreover, considering all the attention that the authors laid on the IID setting, there was no dataset considering it (but, to be clear, I would rather the authors deemphasize the IID setting than an additional focus of experiments on IID data).
    - The baselines being considered are severly lacking. The authors consider ACI (Gibbs & Candès, 2021) and MVP (Bastani et al., 2022). While I wholeheartedly agree that these two methods should be present, they are already a bit dated, and considering only them brings a misleading comparison, as they are not quite the state-of-the-art anymore. Here are some other methods I suggest comparing against:
        - AgACI (https://arxiv.org/abs/2202.07282)
        - Bellman CP (https://arxiv.org/abs/2402.05203)
        - Copula CP (https://arxiv.org/abs/2212.03281)
        - Conformal PID control (https://arxiv.org/abs/2307.16895)
    - Finally, I must note that even with the lacking experimental comparison, I don't exactly see an advantage of the proposed method compared to the baselines.

Additionally, a number of nitpicks (which do not impact my score, but should probably be improved upon):

- Line 073: "without any statistical assumption at all" -- that's not right. While the assumptions are indeed quite lax, there _are_ statistical assumptions: namely, that the protocol described in lines 054-063 is followed (which implies, e.g., that confidence level chosen for a time step is independent of the outcome conditional on the past along with the new covariates).
- On the definition of the quantile in equation (2), you should consider the infimum rather than the minimum to ensure that the quantile is well-defined.
- Line 096: I'd expect a reader of this paper to be familiar with exchangeability. " a relaxation of iid called exchangeability" sounds like talking down. Also, consider citing https://arxiv.org/abs/2005.06095, which is an excellent reference on exchangeability and CP.
- Section 2 should probably be moved to the experiments section, or even the supplementary material. The experiment and the fact that it is embasing has already been mentioned in the introduction; additional motivation is not needed.
- Calling the proposed method 'Bayesian' feels a bit like a misnormer. There is no statistical model being used for Bayesian inference, no likelihood and no Bayes update. It is only related to Bayesian inference by the analogy of regularization being typically equivalent to some form of Bayesian inference and the specification of a 'prior' distribution. That said, again, this is a nitpick and has not weighted into my decision.

**Questions:**

My main gripes with the paper are:

- Too much focus on the IID setting (which is not what online conformal prediction is about)
- Theoretical results for non-IID data being lacking;
- Lacking experimental analysis;
- From the existing experimental analysis, I see no clear advantage of the method, and the theoretical contributions are not notable enough to compensate for it.

(All of them are better described in the 'Weaknesses' section.)

Any clarification or improvement on these points are appreciated.

I believe that, in its current form, the paper is not ready for acceptance, due to the issues raised above, and have thus opted for rejection.
That said, I would be willing to increase my score (and perhaps substantially) should my concerns above be resolved.

---

> ### Author Response · Authors · 2024-11-14
> **Response 1/2**
>
> Thank you very much for your constructive feedback. We will try to run more experiments on real datasets during the rebuttal phase and update here. Meanwhile we'd like to respond to the rest of your major criticisms.
>
> 1. **Invariance to permutation is not a benefit**
>
> You are right, we will remove this "paradox" and only argue about the other one (inconsistency between different $\alpha$). Indeed, we agree that only when the data stream is iid can we confidently say the invariance to permutation is a desirable property.
>
> 2. **Compatibility of performance guarantees with distribution shifts**
>
> We'd like to clarify two possible confusions from your review. The first is on the performance guarantee of Discounted (Theorem 6). Quoting your review,
>
> > And Theorem 6 does not seem to agree with the text surrounding it; the 'optimal' predictions considered for the regret bound seems to be constant over the course of the whole time (since the min over $r$ is outside the sum over time), which is deeply uninteresting. Any minimally decent predictor in the continual distribution shift setting should have predictions that change over time.
>
> Our Theorem 6 bounds the **discounted regret**, which is one of the standard performance measures for "nonstaionary online learning". The intuition is that given a discount factor, it characterizes the performance of the algorithm over a **sliding time window** whose length is determined by the discount factor. Over any time window, the algorithm is compared to the best fixed action specifically for this time window. Importantly, **over different time windows the algorithm will compare to different fixed actions,** which is naturally compatible with the essence of "learning under distribution shifts".
>
> Regarding the example you made:
>
> > A simple case where these things really matter is when your are in an adversarial setting with increasing conformity scores over time. No single prediction is going to be enough, you must increase your intervals over time!
>
> Our discounted algorithm can indeed increase its prediction interval over time. We'll include an experiment to show this.
>
> The second possible confusion is on the utility of the static regret bound (Theorem 2). We'd like to note that **bounding the static regret doesn't mean that we assume the data is IID or exchangeable.** In the learning-theoretic language, the static regret characterizes our algorithm's ability to learn the hypothesis class of time-invariant functions, which is independent of the data generation mechanism. This is also the key motivation for the field of adversarial online learning to focus on the static regret.
>
> 3. **Significance our theoretical results**
>
> Quoting your review,
>
> > The paper's theoretical contributions aren't notable enough nor are any new things possible with the proposed method,
>
> With all due respect, we'd like to push back against this evaluation. Most notably, **our algorithm achieves an adaptive, best-of-both-worlds guarantee that prior works couldn't.** In addition, in the adversarial setting, our algorithm is **the first one to support multiple $\alpha$ queries with monotonicity.**
>
> To be more concrete on the first point, we argue that in many practical applications of CP one has to apply the algorithm without knowing the nature of environment in advance. In such cases it's impossible to say something like "we'll apply Split CP if the data sequence is exchangeable, and ACI otherwise"; instead, **an ideal algorithm needs to work well under all data-generation mechanisms,** which aligns with the statistical concept of *adaptivity*.
>
> Our algorithm does this. Without knowing the nature of the environment beforehand, the same algorithm guarantees that
> - if the data sequence is iid, then the dataset conditional coverage probability is as good as Split CP;
> - over any data sequence, the regret bound of our algorithm is optimal (as good as gradient descent).
>
> We are not aware of any prior work that achieves this.
>
> Besides, we establish the equivalence between a Bayesian algorithm (with uniform prior) and a full-loss FTRL algorithm (with quadratic regularizer), which we see as a substantial technical contribution.
>
> 4. **Experiment**
>
> Finally, the focus of this paper is theoretical, and with our synthetic data experiments (Figure 2 and 3) clearly showing the improvement of our approach over representative baselines, we thought the paper is already quite complete. In addition, we see CP as a particular aspect of "trustworthy computing", where theoretical guarantees are themselves among the key elements that determine the usefulness of an algorithm.

---

> > ### Author Response · Authors · 2024-11-14
> > **Response 2/2**
> >
> > As for experiments on real datasets, we'll try to do more within the rebuttal period. To make this more effective, since you are asking for more datasets and baselines (and unfortunately, there are infinite many...), could you please let us know the specific datasets and baselines you have in mind such that testing on those will improve your evaluation of our paper? We'd like to kindly note that in our experiments (Line 468-478), our algorithm cannot benefit from any extra hyperparameter tuning compared to ACI, which is not the case for some of the baselines suggested in your review (AgACI, Conformal PID). Bellman CP and Copula CP are proposed for the setting of multi-step forecasting, which is different from our focus. We find it quite hard to fairly compare to these baselines in order to obtain scientifically meaningful conclusions.
> >
> > Regarding the results of our experiments, we are not claiming our algorithm beats the baselines in all settings. **What we can confidently say is that it offers concrete improvement when multiple $\alpha$ are queried (Figure 2), while performing similarly as ACI in the conventional setting of a single fixed $\alpha$ (Figure 4 and 5).** So the practitioners of CP are given one more option in their toolbox.
> >
> > Thanks again, and we are looking forward to hearing your thoughts!

---

> > > ### Comment · Reviewer_e4ek · 2024-11-22
> > >
> > > First, I'd like to thank the authors for the thoughtful response.
> > > Continuing the discussion:
> > >
> > > **Regret bounds:** the authors are correct in pointing out that they are, indeed, bounding the more standard notions of regret. It was indeed folly on my part to claim that their results were not informative. While I still think the theorems could be much better, and feel the lack of additional guarantees (for example, the coverage guarantees typical in the online conformal prediction literature; I did not find the authors' explanation in the paper for the lack of one convincing at all), I concede that the method does have some useful guarantees, even if they are presented in a somewhat confusing manner.
> > >
> > > **On the value of the work:** the authors have raised an interesting point: there is indeed much value in an algorithm that is provably valid in iid but is also provably robust to non-iid settings, even if the violations are severe. While I am skeptical that existing methods don't also satisfy this, I am not aware of this being explicitly proven in any of the existing literature, so such guarantees would indeed be of interest. **However**, that is not at all what I gathered by reading the actual paper, and that substantial changes in the writing are probably required for this to change to the direction outlined in the rebuttal. If I'm not mistaken, ICLR this year permits the submission of a revised version of the paper during the discussion period; should the authors submit such a revised version in the direction pointed out in the rebuttal, then I will consider this matter resolved.
> > >
> > > **Invariance to permutations:** removing the supposed 'paradox' is a good step. However, isn't permutation invariance an *undesireable* property for non-iid online conformal prediction? And your method does satisfy it, so isn't this an issue for settings where there are severe violations of exchangeability?
> > >
> > > **Experiments:** In light of the reframing of the paper as a sort of 'simultaneous' guarantee for iid and non-iid data, I agree that less experiments are needed. Nevertheless, I still urge the authors to at least include more baselines, so as to not act as if the field was still as it was in ~2021. Also, regarding the authors' questions:
> > >
> > > - **Datasets:** given the new light given in the rebuttal to the paper, I agree that it is less important to have more datasets. Nevertheless, some suggestions include electricity forecasting (there are some classic datasets for this) and epidemic forecasting (e.g., COVID-19, Dengue, Monkeypox, or other significant outbreaks).
> > > - **Methods:** I don't understand why the lack of benefit from hyperparameter tuning for *your* method is a reason to not compare against those methods. You could even just consider 'default' hyperparameters proposed in the methods' papers. As for Bellman CP and Copula CP, they can still do single-step forecasting, and were shown to do so very well, if I remember correctly -- much better than ACI, in particular.
> > >
> > > Finally, I'd just like to mention that I do not consider the equivalence between a particular Bayesian algorithm with a specific FTRL-type algorithm by itself a substantial contribution. It is welcome, of course, but more as a building block in a larger picture (which is currently not strong).
> > >
> > > Overall, I'm glad that the matter of the meaning of the theorems was resolved, and was mere confusion on my part. There are still some important matters to resolve, though (mainly regarding the writing and presentation of the paper), which hinder me from giving an actual positive score. Neverthless, I'm increasing my score to a borderline reject, and should these points be improved, I will also increase my score accordingly.

---

> ### Author Response · Authors · 2024-11-23
>
> Thanks for your reply. To respond to your main remaining concern (writing doesn't sufficiently emphasize the adaptivity of the algorithm), we revised the paper and updated the pdf. The newly added text is marked in red. In particular, we removed all discussions of permutation invariance, and added a major paragraph to Section 1.2 (summary of results) in order to make the benefit of adaptivity clearer.
>
> We'd also like to kindly remark that besides adaptivity, another major strength of our algorithm is its ability to answer multiple arbitrary confidence level query in a monotonic (or "probabilistically plausible") manner. We hope this strength is also factored into your evaluation. Essentially our paper presents multiple results of possible independent interest (see Section 1.2). People may have different opinion on the ideal proportion of each result, but overall we believe that having multiple results is a plus rather than a minus (Reviewer J1rZ appears to have a more positive sentiment on this).
>
> Appendix D in the updated pdf contains some new experiments. We show that the monotonicity issue demonstrated in Section 2 (synthetic data) also shows up in the stock experiment (actual data). It seems to us that experiments of this type are best suited for our paper. "Multi-confidence consistency" a novel and important evaluation metric that hasn't been considered in the conformal prediction literature before, and it supports our claim well. Overall, we believe that the scientific message in our current experiments is self-complete, solid and clear.
>
> Before reading your latest reply, we also performed the experiment you suggested in your original review (monotonically increasing true score sequence), which demonstrates the effectiveness of discounting. It seems that your criticism regarding discounting is now dismissed, but nonetheless, we included the results in Appendix D for the readers' information.
>
> Finally, we see the role of permutation invariance as an open-ended question. When the true score sequence has temporal structures, being permutation-invariant is not as good as permutation-dependency that "ideally exploits" the temporal structure, but still it's better than permutation-dependency that "wrongly incorporates" the temporal structure. Overall it's hard to draw conclusions at this point, but indeed it suggests a good question for future works.
>
> We hope the above addresses your remaining concerns.

---

> > ### Author Response · Authors · 2024-11-25
> >
> > Dear reviewer,
> >
> > We were wondering if you have any other remaining concern. Thank you for your review and response!

---

> > > ### Comment · Reviewer_e4ek · 2024-12-03
> > >
> > > I have re-reviewed the current state of the paper.
> > >
> > > I think the paper's presentation is not clear. There are multiple ideas that do not mesh very well presented in different portions of the paper, which makes it a very hard read and does not paint a cohesive picture. This was the case, and is even more so now after the revisions. (Also, I am together with reviewer V5nc in beyond bothered by the use of 'Bayesian' throughout the paper. As I said before, I am not factoring this into my score, but I do recommend the authors to reconsider its overwhelming use in the work.)
> > >
> > > I'm also a bit unsettled by the invariance to permutations bit. While the authors raised a reasonable point that "being permutation-invariant is not as good as permutation-dependency that "ideally exploits" the temporal structure, but still it's better than permutation-dependency that "wrongly incorporates" the temporal structure", it is a fairly undersirable property for online CP, and I'm not convinced that other methods 'wrongly incorporate the temporal structure'.
> > >
> > > My qualms with the experiments also remain.
> > >
> > > I thank the reviewers for their effort in the rebuttal, which has significantly raised my perception of the paper since the beginning of the process. However, for the reasons above, I remain with my score of a borderline reject.

---

### Official Review · Reviewer_V5nc · 2024-11-04

**Soundness:** 3
**Presentation:** 3
**Contribution:** 2
**Rating:** 3
**Confidence:** 4

**Summary:**

The authors aim to contribute to the burgeoning literature on conformal prediction, in particular the approach based on quantile tracking.  The main goal of the paper is to develop a method that treats a set of alpha values, and not a single alpha value that has been fixed a priori.  The technical approach is essentially follow-the-regularized leader (FTRL) on the pinball loss, with an interpretation of the regularization in FTRL as a (nonparametric) Bayesian prior.

**Strengths:**

It is true, and perhaps helpful, to point out that quantile-tracking algorithms can be incoherent in terms of the confidence sets that they deliver for different values of alpha.  It is also useful to point out that FTRL can address this issue.  Some of the other detailed critiques of Gibbs & Candes (e.g., the overshooting) are reasonable.

**Weaknesses:**

I'm not entirely sure what I've learned from the paper, other than being reminded of some of the virtues of FTRL.

Indeed, I want to emphasize that as best I can tell, this is a FTRL paper.  The connection to Bayesian inference is weak at best, and not very helpful.  There's a regularizer, but that alone doesn't make for Bayesian inference.  At the end of the paper, there's a suggestion that when regularization involves adding a uniform distribution to an empirical distribution we can interpret this as the (posterior mean) of a Dirichlet process, but I'm at a loss as to why that kind of Bayesian nonparametric terminology is called for here.  Note in particular that a Bayesian would consider the observations forming the empirical distribution to be iid draws from a multinomial, quite out of the spirit of the "no statistical assumptions" statement that the authors make.  Moreover, if one is really wanting to be Bayesian here, then it's the entire quantile process that should be the object of inference.

The main motivation of the paper doesn't seem to be Bayesian at all, but rather simply online learning, in a setting in which quantiles corresponding to multiple alpha values are desired.  Now, I'm not sure exactly when and how multiple alpha values might be desired by a downstream user.  Is there a decision-theoretic justification for asking for this?  The authors don't go much beyond simply suggesting that it's desirable in some intuitive sense.

There are frameworks in statistics for making repeated tentative decisions over time, e.g., the nonnegative supermartingale
approach, and such decision-theoretic frameworks are clearer (to me) ways of getting at the issue being hinted at here.

The other motivation for pursuing this approach is the "paradox" (lack of invariance to permutations) associated with quantile tracking.
But this critique seems to lose its punch when one remembers that one of the main motivations for the quantile tracking approaches
is that of handling nonstationarity.

**Questions:**

My main suggestion is to be clearer on what exactly a downstream user might be asking for, in specific inferential use cases, to support the goal of tracking with respect to multiple alpha values simultaneously.  A secondary suggestion is to clarify in what sense this is really Bayesian inference, beyond simply having a sum-of-distributions interpretation.  Why isn't this just an FTRL story pure and simple.

---

> ### Author Response · Authors · 2024-11-14
>
> Thank you very much for your constructive review. Regarding your comments,
>
> 1. **Reason to call our algorithm Bayesian**
>
> To begin with, we thought it's quite important to note that **we only call our algorithm Bayesian, rather than the problem setting and the analysis.**
>
> **From only the algorithmic perspective,** as discussed in Section 5, our belief update backbone is the same as the standard *Bayesian distribution estimator*. This estimator doesn't seem have a separate fancy name from the Bayesian statistics literature, but it should be a fairly standard textbook-level concept as covered by (Chapter 23, Gelman et al., Bayesian Data Analysis, 2021). We see a major novelty of our work as showing the **equivalence** of a *Bayesian algorithm* from statistics and a *full-loss FTRL algorithm* from adversarial online learning, which is significant but hasn't been concretely explored in the past.
>
> **As for the problem setting and the analysis,** ours are indeed very different from Bayesian statistics, as also pointed out by your review:
>
> > Note in particular that a Bayesian would consider the observations forming the empirical distribution to be iid draws from a multinomial, quite out of the spirit of the "no statistical assumptions" statement that the authors make.
>
> Indeed, traditional Bayesian statistics doesn't consider the "adversarial settings" like in our work. This leads to the main **operational value of the above algorithmic equivalence:** we could give certain performance guarantees of the proposed Bayesian algorithm, without statistical assumptions at all. We call such analytical framework **"adversarial Bayes"**.
>
> 2. **Importance of multiple confidence level queries**
>
> Supporting multiple $\alpha$ queries can be useful in plenty of applications.
>
> - Imagine a "central" CP algorithm predicting confidence sets on the stock prices, and different downstream users apply those sets for their respective investment activities. Each user has a different risk preference, which determines a different $\alpha$ to be queried from the CP algorithm.
> - Another case is when a single downstream user applies the confidence sets in an iterative subroutine, which requires multiple queries in the same round. For example, the user first queries $\alpha=90\%$ and gets the algorithm's prediction set. If the set is deemed "too large" then the user might consider lowering $\alpha$ to 80% and query again (in the same round). So on and so forth.
>
> 3. **Invariance to permutation is not a benefit**
>
> This is a good point! We will remove this "paradox" and only argue about the other one (inconsistency between different $\alpha$). Indeed, we agree that only when the data stream is iid can we confidently say the invariance to permutation is a desirable property.
>
> 4. **Why not a pure and simple FTRL story?**
>
> Compared to a pure FTRL story, there are two benefits of the Bayesian statistical perspective:
> - **Computation.** It's worth-noting that we use the "full-loss" FTRL rather than the linearized version of FTRL more commonly found in the generic online learning literature. Such unpopularity of full-loss FTRL has been mainly due to computational reasons, as the online learning algorithm needs to solve a convex program in each round, similar to the RHS of Eq.(6).
> In contrast, our paper uses the Bayesian interpretation to **avoid the need of convex programs** (the special structure of CP is important here), and the algorithm admits a backbone-head decomposition which **enables multiple $\alpha$ queries in a computationally efficient manner.**
> - **Probabilistic bound under IID assumption.** CP is conventionally a statistics problem, and the Bayesian perspective allows us to obtain a **coverage probability bound** assuming IID (Section 3.2), which is beyond the scope of typical regret analyses on FTRL. In particular, the proofs of Theorem 4 and 5 are essentially built on this Bayesian perspective.
>
> Besides these concrete operational benefits, we would argue that connecting full-loss FTRL with Bayesian statistics is itself quite interesting from a technical perspective. It requires exploiting the structure of the quantile losses, which hasn't been considered in the generic online learning literature. At least we found it surprising that the quadratic regularizer shows up when the prior is uniform. Section 1.2 summarizes the conceptual takeaway of our paper and the associated quantitative benefits.
>
> Please let us know if you have any remaining concern.

---

> > ### Author Response · Authors · 2024-11-25
> >
> > Dear reviewer,
> >
> > Since the rebuttal period will end soon, we were wondering if our response above has changed your opinion? We are happy to address any remaining concern. Thank you!

---

> > > ### Comment · Reviewer_V5nc · 2024-11-29
> > >
> > > I'm not convinced.  Indeed, the answers heightened my concerns.  One of my queries
> > > was regarding the need for having multiple alpha values (confidence levels).  I wanted
> > > to hear what exactly the downstream user might want to do with multiple alpha values.
> > > The authors responded by saying "the user first queries and gets the algorithm's prediction
> > > set. If the set is deemed "too large" then the user might consider lowering to 80% and
> > > query again (in the same round). So on and so forth."  But this is a form of p-hacking!
> > > There is no guarantee regarding the overall confidence level if one looks at one level
> > > and then moves to another.
> > >
> > > Second, I continue to believe that this is just an application of Follow the Regularized
> > > Leader (FTRL).  I really don't see any significant connection to Bayesian thinking.  The authors
> > > say "we only call our algorithm Bayesian, rather than the problem setting and the analysis".
> > > But without the "problem setting", which is an inferential problem and an appropriate prior
> > > and likelihood to define an appropriate posterior for inference, the Bayesian perspective
> > > isn't contributing anything.  In particular, there really isn't a well-defined notion of a
> > > "Bayesian algorithm".  Moreover, the authors say "we could give certain performance guarantees
> > > of the proposed Bayesian algorithm, without statistical assumptions at all. We call such
> > > analytical framework "adversarial Bayes".  Sorry, but this is just language.  There is no
> > > "analytical framework" that is Bayesian and applies to arbitrary sequences without statistical
> > > assumptions.  The current paper certainly does not provide any such framework.
> > >
> > > Finally, a statement like "the Bayesian perspective allows us to obtain a coverage probability
> > > bound assuming IID (Section 3.2), which is beyond the scope of typical regret analyses on FTRL."
> > > is, again, simply words.  Bayesian perspectives provide credible intervals; they don't provide
> > > coverage.  And bringing in IID for a regret analysis makes no sense.
> > >
> > > Sorry, but there's too much that's conceptually mixed up here for me to support acceptance
> > > for this paper.

---

> > > > ### Author Response · Authors · 2024-11-29
> > > >
> > > > Thank you for following up. We respect your evaluation, but there are certain major misunderstandings in your review.
> > > >
> > > > Regarding the motivation of multiple confidence level queries, we were wondering if you find the other case in our response convincing? The conformal predictor can be used by many users, and each user can query a different confidence level due to a different risk preference.
> > > >
> > > > On calling our *algorithm* "Bayesian", we additionally note that in the adversarial setting one could still pretend the nature is iid and apply the Bayes' theorem. As shown in Section 5, our belief update Eq.(5) is equivalent to the posterior mean with a Dirichlet process prior. Our work analyzes the same algorithm, but without the iid assumption (for the most parts).
> > > >
> > > > To provide another perspective, Xu and Zeevi (2023) designed Bayesian algorithms for adversarial bandit problems, where the use of the word "Bayesian" is similar to our work. This is one of the outstanding paper awards at ICML 2023, which means that the same terminology is likely acceptable to the majority of the field.
> > > >
> > > > Finally, we note that we were not "bringing in IID for a regret analysis". Our coverage probability guarantee (Theorem 4) is independent of our regret analysis, and it comes very concretely from the Bayesian interpretation of full-loss FTRL. We also have a separate "online-to-batch" bound for the *excess quantile risk*, Theorem 5.
> > > >
> > > > We would sincerely appreciate other reviewers and program committee members to chime in.
> > > >
> > > > **References**
> > > >
> > > > Yunbei Xu and Assaf Zeevi. Bayesian design principles for frequentist sequential learning. In International Conference on Machine Learning, pp. 38768–38800. PMLR, 2023.

---

### Official Review · Reviewer_J1rZ · 2024-11-04

**Soundness:** 4
**Presentation:** 4
**Contribution:** 4
**Rating:** 8
**Confidence:** 3

**Summary:**

The paper  presents a  novel approach to online conformal prediction (CP) that uses Bayesian regularization to overcome the limitations of previous direct and indirect CP methods. Conventional methods rely either on empirical quantile predictions (direct approach) or on adversarial optimization (indirect approach).  The Bayesian approach presented here addresses these issues by introducing a regularized Bayesian framework that combines empirical and prior-based distributions, thereby enabling more robust predictions without statistical assumptions about the data distribution.

Key contributions include:
- by implementing Bayesian regularization, the model provides reliable confidence sets that avoid the validity problems of purely adversarial CP methods, such as non-monotonicity across confidence levels.
- The Bayesian CP model achieves an optimal regret boundary, maintains competitive performance across different confidence levels, and provides stable results in both iid and adversarial contexts.
- Unlike traditional methods that require a single preset confidence level, the Bayesian approach supports online responses to arbitrary confidence queries - a critical improvement for dynamic, real-world applications.

The framework is theoretically validated with regret bounds and empirically demonstrated on both synthetic and real-world datasets, showing superior adaptability and efficiency compared to existing CP methods.

**Strengths:**

- The paper is clearly written and comprehensively explains its novel Bayesian approach to online conformal prediction. It effectively outlines the limitations of existing methods and clearly presents its contributions. The theoretical foundations are well-developed, and the empirical results are presented to complement the theory, making the concepts accessible to readers. The structure allows for an easy understanding of the core ideas, technical details, and the practical advantages of the proposed method.
- The paper contains many interesting results, which are novel to my best knowledge
    - Theorem 2 establishes the regret bound for the Bayesian CP algorithm, showing that it achieves \( O(R\sqrt{T}) \) regret for any sequence length \( T \) and confidence level \( \alpha \) in adversarial settings.
    - Theorem 3 introduces the quantized version of the algorithm, demonstrating that it achieves the same \( O(R\sqrt{T}) \) regret bound with reduced memory requirements of \( O(\sqrt{T}) \).
   -  Theorem 4 shows, under the iid assumption,  that the Bayesian CP algorithm achieves probabilistic coverage guarantees similar to ERM-based CP in iid contexts.
   - Theorem 5  addresses the excess quantile risk under the iid assumption, demonstrating that the Bayesian CP algorithm achieves similar performance as traditional ERM approaches in iid environments.
   - Theorem 6 provides a discounted regret bound for the variant of the Bayesian CP algorithm designed to handle continual distribution shifts. This bound matches minimax optimality, indicating the algorithm’s resilience in non-stationary environments.

**Weaknesses:**

I do not see serious weak points.

**Questions:**

- Can you provide more details on MultiOGD [in Section 2]. The current version is so sketchy that it is difficult to read.
- Provide more details in Section 3 [I know that the paper is short, you could provide more details in the appendix for example]

---

> ### Author Response · Authors · 2024-11-14
>
> Thank you for your careful review that found our paper's strengths. We really appreciate that!
>
> Regarding your suggestions, we will try to add additional clarifying text to Section 3 in the revision. As for the MultiOGD baseline:
>
> - First, recall that OGD itself has Eq.(3) in our paper as its update rule, which requires a fixed confidence level $\alpha$ throughout its operation (notice that $\alpha$ is used to define the quantile loss functions).
> - MultiOGD is perhaps the simplest extension of OGD that doesn't require fixing $\alpha$ beforehand. To do that, MultiOGD maintains a bunch of *independent* OGD algorithms (called base algorithms), each targeting a different confidence level $\tilde \alpha$. At the beginning of each round, all these base algorithms (with different $\tilde \alpha$) would have their predictions ready. If a user queries a specific confidence level $\alpha$, then MultiOGD will output the prediciton of the base algorithm whose $\tilde \alpha$ value is the closest to $\alpha$.
> - Particularly in Section 2, we only consider two different users in the experiments, each querying a fixed $\alpha$ value. In this case MultiOGD just becomes OGD itself, and our experiments essentially show that "two copies of OGD with different $\alpha$ can be inconsistent with each other".
>
> Thanks again, and please let us know any remaining question!

---

> > ### Comment · Reviewer_J1rZ · 2024-11-26
> >
> > I have read your answers: Of course the paper is not perfect, but it is a challenging and novel topic, and it is often more difficult to lay the initial groundwork than to add the finishing touches. I remain favorably impressed by the results presented in this paper... However, I am not an expert in the online approach, so my assessment is probably more generous than that of the other reviewers, if I understand their criticisms correctly.

---

> > > ### Author Response · Authors · 2024-11-26
> > >
> > > Thank you again for your comments!

---

### Official Review · Reviewer_PGHS · 2024-11-05

**Soundness:** 3
**Presentation:** 3
**Contribution:** 2
**Rating:** 6
**Confidence:** 3

**Summary:**

This paper introduces an algorithm to compute prediction sets in the context of online conformal prediction : given a sequence of previous input and labels, the goal is to compute (for a new observed input) a score threshold leading to a prediction set.
The main idea of their algorithm is to define, at each step, a mixture of a prior distribution P_0 and the empirical distribution of the previous scores, and to use the quantile of this mixture as score thresholds. This approach is similar to the Follow The Regularized Leader (FTRL) in the bandits literature. From this similarity, the authors compute a regret bound for their algorithm and show that for a simple choice of prior (uniform distribution), their algorithm has a regret of O(\sqrt{T}). In the case of distribution shift, a variant of their algorithm reaches constant discounted regret.
Moreover, contrary to other existing approaches, the authors claim that the prediction intervals computed using their method satisfy a form of monoticity (meaning that the score thresholds for a higher coverage will be higher than those  for a lower coverage).
Numerical experiments show that the method performs competitively with other approaches (online gradient descent, ERM, etc.)

**Strengths:**

The paper is overall clearly written (I did not check carefully the math in Appendix B) and the proposed method is both simple and seems to have good guarantees.

**Weaknesses:**

I would have like more details on the numerical experiments. From what I understand, the figures only show that the method reaches the target coverage, but what about the size of the prediction sets that it produces ? How does it compare to other methods ? Also, the memory usage of the method ( O(\sqrt(T}) for the quantized version) seems redhibitory at first glance, but is not evaluated in practice in the experiments.

**Questions:**

- In general in Bayesian methods for UQ, the role of the prior is very important to have good performance. So I would be curious to know if other reasonable choices of priors for the score thresholds would improve the regret compared to the uniform prior (apart from the one mentioned in line 320). Is this something that you have explored ?

---

> ### Author Response · Authors · 2024-11-14
>
> Thank you very much for your review. Regarding your questions:
>
> 1. **"what about the size of the prediction sets (in the experiment)?"**
>
> Figure 5 in the paper shows the score thresholds predicted by our algorithm and the baselines, which essentially determines the sizes of the prediction sets. It could be visually seen that all three algorithms perform similarly in terms of the set sizes.
>
> 2. **"the $O(\sqrt{T})$ memory usage seems redhibitory at first glance, but is not evaluated in practice in the experiments."**
>
> In principle, our algorithm needs to maintain a histogram of the past true scores, and the memory usage is determined by the number of bins the histogram contains. We need $O(\sqrt{T})$ bins for theoretical purposes, but in practice it's natural to keep the number of bins constant (in our experiment we pick 40, which induces negligible approximation error). This is the same reasoning as in (Bastani et al., 2022) which is also built on binning.
>
> 3. **The role of prior**
>
> Theoretically speaking the uniform prior is the "minimax optimal" choice: for any other prior, there exists an environment where it's worse than the uniform. But in practice we would expect better performance using suitable non-uniform priors, just like typical Bayesian algorithms. Appendix A contains a comparison of our approach to Bayesian UQ.
>
> We hope the above answers your questions!

---

> > ### Comment · Reviewer_PGHS · 2024-11-26
> >
> > Thanks to the authors for answering my questions. I decide to keep my score and confidence unchanged.

---

### Meta-Review · Area_Chair_Vscf · 2024-12-19

**Metareview:**

Summary:

This paper introduces a Bayesian-inspired approach for online conformal prediction that aims to integrate strengths from both "data-centric" and "iterate-centric" methodologies. While it has theoretical merit, significant issues remain unresolved regarding its conceptual clarity, novelty, and experimental evaluation.

Strengths:

The proposal addresses the inconsistency of predictions for different confidence levels in traditional conformal prediction approaches.
Theoretical guarantees are provided, including regret bounds, which are well-aligned with the online learning framework.

Weaknesses:

Use of Bayesian Terminology: The use of "Bayesian" in the title and throughout the paper is misleading. The method lacks a clear Bayesian inferential framework, as highlighted by multiple reviewers. This terminology seems primarily rhetorical, which undermines the clarity and positioning of the work.

Lack of Novelty: The core method appears to be a straightforward application of Follow the Regularized Leader (FTRL). The claimed contributions, including the use of priors and a "Bayesian perspective," fail to substantiate meaningful technical advancements.

Weak Experimental Validation: The experiments are limited in scope, lack comparisons with state-of-the-art baselines, and fail to convincingly demonstrate practical advantages over existing methods. The focus on stock market data and the absence of broader, diverse datasets reduce the impact of the empirical results.

Conceptual Ambiguity: The discussion fails to justify the practical necessity of multiple alpha values, and the proposed approach risks enabling practices akin to p-hacking. Moreover, the invariance to permutation is poorly motivated and arguably undesirable in online settings.

Discussion:
While some reviewers found the idea of combining "data-centric" and "iterate-centric" approaches interesting, the conceptual and methodological execution is not sufficiently robust. Concerns regarding the misleading use of Bayesian terminology and the lack of novelty relative to FTRL were not adequately addressed in the discussion. Furthermore, the weak experimental evaluation fails to compensate for these conceptual shortcomings.

Conclusion:
Although the paper touches on an interesting topic, its issues—particularly the lack of conceptual clarity and convincing novelty—prevent me from recommending acceptance at this stage. Future work should focus on refining the core methodology, clarifying its conceptual underpinnings, and providing more rigorous experimental validation.

**Additional Comments On Reviewer Discussion:**

See above.

---

### Decision · Program_Chairs · 2025-01-22

Reject